# One More Step Towards Reality: Cooperative Bandits with Imperfect Communication

**Udari Madhushani**
Princeton University
udarim@princeton.edu

**Abhimanyu Dubey**
Massachusetts Institute of Technology
dubeya@mit.edu

**Naomi Ehrich Leonard**
Princeton University
naomi@princeton.edu

**Alex Pentland**
Massachusetts Institute of Technology
pentland@mit.edu

## Abstract

The cooperative bandit problem is increasingly becoming relevant due to its applications in large-scale decision-making. However, most research for this problem focuses exclusively on the setting with perfect communication, whereas in most real-world distributed settings, communication is often over stochastic networks, with arbitrary corruptions and delays. In this paper, we study cooperative bandit learning under three typical real-world communication scenarios, namely, (a) message-passing over stochastic time-varying networks, (b) instantaneous reward-sharing over a network with random delays, and (c) message-passing with adversarially corrupted rewards, including byzantine communication. For each of these environments, we propose decentralized algorithms that achieve competitive performance, along with near-optimal guarantees on the incurred group regret as well. Furthermore, in the setting with perfect communication, we present an improved delayed-update algorithm that outperforms the existing state-of-the-art on various network topologies. Finally, we present tight network-dependent minimax lower bounds on the group regret. Our proposed algorithms are straightforward to implement and obtain competitive empirical performance.

## 1 Introduction

The cooperative multi-armed bandit problem involves a group of $N$ agents collectively solving a multi-armed bandit while communicating with one another. This problem is relevant for a variety of applications that involve decentralized decision-making, for example, in distributed controls and robotics [31] and communication [21]. In the typical formulation of this problem, a group of agents are arranged in a network $G = (\mathcal{V}, \mathcal{E})$, wherein each agent interacts with the bandit, and communicates with its neighbors in $G$, to maximize the cumulative reward.

A large body of recent work on this problem assumes the communication network $G$ to be fixed [20, 22]. Furthermore, these algorithms inherently require precise communication, as they construct careful confidence intervals for cumulative arm statistics across agents, e.g., for stochastic bandits, it has been shown that the standard UCB1 algorithm [3] with a *neighborhood* confidence interval is close to optimal [11, 20], and correspondingly, for adversarial bandits, a neighborhood-weighted loss estimator can be utilized with the EXP3 algorithm to provide competitive regret [7]. Such approaches are indeed feasible when communication is perfect, e.g., the network $G$ is fixed, and messages are not lost or corrupted. In real-world environments, however, this is rarely true: messages can be lost, agents can be byzantine, and communication networks are rarely static [23]. This aspect has hence received much attention in the distributed optimization literature [35]. However, contrary to network optimization where dynamics in communication can behave synergistically [17], bandit problems

additionally bring a decision-making component requiring an explore-exploit trade-off. As a result, external randomness and corruption are incompatible with the default optimal approaches, and require careful consideration [33, 25]. This motivates us to study the multi-agent bandit problem under real-world communication, which regularly exhibits external randomness, delays and corruptions. Our key contributions include the following.

**Contributions**. We provide a set of algorithms titled Robust Communication Learning (RCL) for the cooperative stochastic bandit under three real-world communication scenarios.

First, we study stochastic communication, where the communication network $G$ is time-varying, with each edge being present in $G$ with an unknown probability $p$. For this setting, we present a UCB-like algorithm, RCL-LF (Link Failures), that directs agent $i$ to discard messages with an additional probability of $1 - p_i$ in order to control the bias in the (stochastic) reward estimates. RCL-LF obtains a group regret of $\mathcal{O}\left(\left(\sum_{i=1}^{N}(1 - p \cdot p_i) + \sum_{\mathcal{C} \in \mathcal{C}}(\max_{i \leq \mathcal{C}} p_i) \cdot p\right)\left(\sum_{k=1}^{K} \frac{\log T}{\Delta_k}\right)\right)$, where $\mathcal{C}$ is a non overlapping clique covering of $G$, $T$ is time horizon, and $\Delta_k$ is the difference in reward mean between the optimal and $k$th arm. The regret exhibits a smooth interpolation between known rates for no communication ($p = 0$) and perfect communication ($p = 1$).

Second, we study the case where messages from any agent can be delayed by a random (but bounded) number of trials $\tau$ with expectation $\mathbb{E}[\tau]$. For this setting, simple reward-sharing with a natural extension of the UCB algorithm (RCL-SD (Stochastic Delays)) obtains a regret of $\mathcal{O}\left(\bar{\chi}(G) \cdot \left(\sum_{k>1} \frac{\log T}{\Delta_k}\right) + \left(N \cdot \mathbb{E}[\tau] + \log(T) + \sqrt{N \cdot \mathbb{E}[\tau] \log(T)}\right) \cdot \sum_{k>1} \Delta_k\right)$, which is reminiscent of that of single-agent bandits with delays [18] (Remark 4). Here $\bar{\chi}(G)$ is the clique covering number of $G$.

Third, we study the *corrupted* setting, where any message can be (perhaps in a byzantine manner) corrupted by an unknown (but bounded) amount $\epsilon$. This setting presents the two-fold challenge of receiving feedback after (variable) delays as well as adversarial corruptions, making existing arm elimination [25, 8, 16] or cooperative estimation [11] methods inapplicable. We present algorithm RCL-AC (Adversarial Corruptions) that overcomes this issue by limiting exploration only to well-positioned agents in $G$, who explore using a *hybrid* robust arm elimination and local confidence bound approach. RCL-AC obtains a regret of $\mathcal{O}\left(\psi(G_\gamma) \cdot \sum_{k=1}^{K} \frac{\log T}{\Delta_k} + N \sum_{k=1}^{K} \frac{\log \log T}{\Delta_k} + NTK\gamma\epsilon\right)$, where $\psi(G_\gamma)$ denotes the domination number of the $\gamma$ graph power of $G$, which matches the rates obtained for corrupted single-agent bandits without knowledge of $\epsilon$.

Finally, for perfect communication, we present a simple modification of cooperative UCB1 that provides significant empirical improvements, and also provides minimax lower bounds on the group regret of algorithms based on message-passing.

**Related Work.** A variant of the networked adversarial bandit problem without communication constraints (e.g., delay, corruption) was studied first in the work of Awerbuch and Kleinberg [4], who demonstrated an average regret bound of order $\sqrt{(1 + K/N)T}$. This line of inquiry was generalized to networked communication with at most $\gamma$ rounds of delays in the work of [7], that demonstrate an average regret of order $\sqrt{(\gamma + \alpha(G_\gamma)/N)KT}$ where $\alpha(G_\gamma)$ denotes the independence number of $G_\gamma$, the $\gamma$-power of network graph $G$. This line of inquiry has been complemented for the stochastic setting with problem-dependent analyses in the work of Kolla et al. [20] and Dubey and Pentland [11]. The former presents a UCB1-style algorithm with instantaneous reward-sharing that obtains a regret bound of $\mathcal{O}(\alpha(G) \cdot \sum_{k=1}^{K} \frac{\log T}{\Delta_k})$ that was generalized to message-passing communication with delays in the latter.

Alternatively, Landgren et al. [22] consider the multi-agent bandit where communication is done instead using a *running consensus* protocol, where neighboring agents average their reward estimates using the DeGroot consensus model [10]. This algorithm was refined in the work of Martínez-Rubio et al. [29] by a delayed mixing scheme that reduces the bias in the consensus reward estimates. A specific setting of Huber contaminated communication was explored in the work of Dubey and Pentland [12]; however, in contrast to our algorithms, that work assumes that the total contamination likelihood is known *a priori*. Additionally, multi-agent networked bandits with stochastic communication was considered in Madhushani and Leonard [26, 27, 28], however, only for regular networks and multi-star networks.

Table 1: Quantity (with notation) for any graph $G$.

| Average degree ($d$) | Maximum degree ($d_{\max}$) | Degree of $i$ ($d_i$) | Independence number ($\alpha$) |
| --- | --- | --- | --- |
| Message life ($\gamma$) | Minimum degree ($d_{\min}$) | Neighborhood of $i$ ($\mathcal{N}_i$) | Domination number ($\psi$) |
| $k$-power of $G$ ($G_k$) | Diameter ($d_\star$) | $\mathcal{N}_i \cup \{i\}$ ($\mathcal{N}_i^+$) | Clique covering number ($\bar{\chi}$) |

Our work also relates to aspects of stochastic delayed feedback and corruptions in the context of single-agent multi-armed bandits. There has been considerable research in these areas, beginning from the early work of Weinberger and Ordentlich [34] that proposes running multiple bandit algorithms in parallel to account for (fixed) delayed feedback. Vernade et al. [33] discuss the multi-armed bandit with stochastic delays, and provide algorithms using optimism indices based on the UCB1 [3] and KL-UCB [13] approaches. Stochastic bandits with adversarial corruptions have also received significant attention recently. Lykouris et al. [25] present an arm elimination algorithm that provides a regret that scales linearly with the total amount of corruption, and present lower bounds demonstrating that the linear dependence is inevitable. This was followed up by Gupta et al. [15] who introduce the algorithm BARBAR that improves the dependence on the corruption level by a better sampling of worse arms. Alternatively, Altschuler et al. [1] discuss best-arm identification under contamination, which is a weaker adversary compared to the one discussed in this paper. The corrupted setting discussed in our paper combines both issues of (variable) delayed feedback along with adversarial corruptions, and hence requires a novel approach.

In another line of related work, Chawla *et al.*[8] discuss gossip-based communication protocols for cooperative multi-armed bandits. While the paper provides similar results, there are several differences in the setup considered in Chawla et al compared to our setup. First, we can see that Chawla *et al.*do not provide a uniform $\mathcal{O}(\frac{1}{N})$ speedup, but in fact, their regret depends on the difficulty of the first $\frac{K}{N}$ arms, which is a $\mathcal{O}(\frac{1}{N})$ speed up only when all arms are "uniformly" suboptimal, i.e., $\Delta_i \approx \Delta_j \forall i, j \in [K]$. In contrast, our algorithm will always provide a speed up of order $\frac{\alpha(G_\gamma)}{N}$ regardless of the arms themselves, and when we run our algorithm by setting the delay parameter $\gamma = d_\star(G)$ (diameter of the graph $G$), we obtain an $\mathcal{O}(\frac{1}{N})$ speedup regardless of the sparsity of $G$. Additionally, our constants (per-agent) scale as $\mathcal{O}(K)$ in the worst case, whereas Chawla et al obtain a constant between $\mathcal{O}(K + (\log N)^\beta)$ and $\mathcal{O}(K + N^\beta)$ for some $\beta \gg 1$, based on the graph structure, which can dominate the $\log T$ term when we have a large number of agents present.

## 2 Preliminaries

**Notation (Table 1).** We denote the set $a, ..., b$ as $[a, b]$, and as $[b]$ when $a = 1$. We define the indicator of a Boolean predicate $x$ as $\mathbf{1}\{x\}$. For any graph $G$ with diameter $d_\star(G)$, and any $1 \leq \gamma \leq d_\star(G)$, we define $G_\gamma$ as the $\gamma$-power of $G$, i.e., the graph with edge $(i, j)$ if $i, j$ are at most a distance $\gamma$.

**Problem Setting**. We consider the cooperative stochastic multi-armed bandit problem with $K$ arms and a group $\mathcal{V}$ of $N$ agents. In each round $t \in [T]$, each agent $i \in \mathcal{V}$ pulls an arm $A_i(t) \in [K]$ and receives a random reward $X_i(t)$ (realized as $r_i(t)$) drawn i.i.d. from the corresponding arm's distribution. We assume that each reward distribution is sub-Gaussian with an unknown mean $\mu_k$ and unknown variance proxy $\sigma_k^2$ upper bounded by a known constant $\sigma^2$. Without loss of generality we assume that $\mu_1 \geq \mu_2 \ldots \geq \mu_K$ and define $\Delta_k := \mu_1 - \mu_k, \forall k > 1$, to be the reward gap (in expectation) of arm $k$. Let $\overline{\Delta} := \min_{k>1} \Delta_k$ be the minimum expected reward gap. For brevity in our theoretical results, we define $g(\xi, \sigma) := 8(\xi + 1)\sigma^2 = o(1)$ and $f(M, G) := M \sum_{k>1} \Delta_k + 4 \sum_{i=1}^N (3 \log(3(d_i(G) + 1)) + (\log (d_i(G) + 1))) \cdot \sum_{k>1} \Delta_k = o((M + N \log N) \cdot \sum_{k>1} \Delta_k)$.

**Networked Communication (Figure 1).** Let $G = (\mathcal{V}, \mathcal{E})$ be a connected, undirected graph encoding the communication network, where $\mathcal{E}$ contains an edge $(i, j)$ if agents $i$ and $j$ can communicate directly via messages with each other. After each round $t$, each agent $j$ broadcasts a message $\boldsymbol{m}_j(t)$ to all their neighbors. Each message is forwarded at most $\gamma$ times through $G$, after which it is discarded. For any value of $\gamma > 1$, the protocol is called *message-passing* [24], but for $\gamma = 1$ it is called *instantaneous reward sharing*, as this setting has no delays in communication.

**Exploration Strategy (Figure 2).** For Sections 3 and 4 we use a natural extension of the UCB1 algorithm for exploration. Thus we modify UCB1 [3] such that at each time step $t$ for each arm $k$ each agent $i$ constructs an upper confidence bound, i.e., the sum of its estimated expected reward

2. Adds $\boldsymbol{m}_i(t)$ to the set of messages $\mathbf{M}_i(t)$, broadcasts all messages in $\mathbf{M}_i(t)$ to its neighbors and receives messages $\mathbf{M}'_i(t)$ from its neighbors.

3. Computes $\mathbf{M}_i(t+1)$ from $\mathbf{M}'_i(t)$ by discarding all messages sent prior to round $t - \gamma$.

This is called *instantaneous reward sharing* for $\gamma = 1$ (no delays), and *message-passing* for $\gamma > 1$.

Figure 1: The cooperative bandit protocol with delay parameter $\gamma$.

---

**For** $t = 1, 2, ...,$ **each agent** $i \in \mathcal{V}$

1. Calculates, for each arm $k \in [K]$, $Q^i_k(t-1) = \widehat{\mu}^i_k(t-1) + \sigma \sqrt{\frac{2(\xi+1)\log(t-1)}{N^i_k(t-1)}}$, where $N^i_k(t-1)$ is the number of reward samples available for arm $k$ at time $t$.

2. Plays arm $A_i(t) = \arg\max_k Q^i_k(t-1)$

---

Figure 2: Cooperative UCB1 which uses additional arm pulls from messages.

$\widehat{\mu}^i_k(t-1)$ (empirical average of all the observed rewards) and the uncertainty associated with the estimate $C^i_k(t-1) := \sigma \sqrt{\frac{2(\xi+1)\log t}{N^i_k(t-1)}}$ where $\xi > 1$, and pulls the arm with the highest bound.

**Regret.** The performance measure we consider, *group regret*, is a straightforward extension of *pseudo regret* for a single agent. Group regret is the regret (in expectation) incurred by the group $\mathcal{V}$ by pulling suboptimal arms. The group regret is given by $\mathsf{Reg}_G(T) = \sum_{i=1}^N \sum_{k>1} \Delta_k \cdot \mathbb{E}\left[n^i_k(t)\right]$, where $n^i_k(t)$ is the number of times agent $i$ pulls the suboptimal arm $k$ up to (and including) round $t$.

Before presenting our algorithms and regret upper bounds we present some graph terminology.

**Definition 1** (Clique covering number). A clique cover $\mathcal{C}$ of any graph $G = (\mathcal{V}, \mathcal{E})$ is a partition of $\mathcal{V}$ into subgraphs $C \in \mathcal{C}$ such that each subgraph $C$ is fully connected, i.e., a clique. The size of the smallest possible covering $\mathcal{C}^\star$ is known as the *clique covering number* $\bar{\chi}(G)$.

**Definition 2** (Independence number). The *independence number* $\alpha(G)$ of $G = (\mathcal{V}, \mathcal{E})$ is the size of the largest subset of $\mathcal{V}_\alpha \subseteq \mathcal{V}$ such that no two vertices in $\mathcal{V}_\alpha$ are connected.

**Definition 3** (Domination number). The *domination number* $\psi(G)$ of $G = (\mathcal{V}, \mathcal{E})$ is the size of the smallest subset $\mathcal{V}_\psi \subseteq \mathcal{V}$ such that each vertex not in $\mathcal{V}_\psi$ is adjacent to at least one agent in $\mathcal{V}_\psi$.

**Organization.** In this paper, we study three specific forms of communication errors. Section 3 discusses the case when, for both *message-passing* and *instantaneous reward-sharing*, any message forwarding fails independently with probability $p$, resulting in stochastic communication failures. Section 4 discusses the case when *instantaneous reward-sharing* incurs a random (but bounded) delay. Section 5 discusses the case when the outgoing reward from any message may be corrupted by an adversarial amount at most $\epsilon$. Finally, in Section 6, we discuss an improved algorithm for the case with perfect communication and present minimax lower bounds on the problem. We present all proofs in the Appendix and present proof-sketches highlighting the central ideas in the main paper.

## 3 Probabilistic Message Selection for Random Communication Failures

The fundamental advantage of cooperative estimation is the ability to leverage observations about suboptimal arms from neighboring agents to reduce exploration. However, when agents are communicating over an arbitrary graph, the amount of information an agent receives varies according to its connectivity in $G$. For example, agents with a large number of neighbors receive more information, leading them to begin exploitation earlier than agents with fewer neighbors. This means that well-connected agents exhibit better performance early on, but because they quickly do only exploiting, agents that are poorly connected typically only observe exploitative arm pulls, which requires them to explore for longer in order to obtain similarly good estimates for suboptimal arms, increasing their regret. The disparity between performance in well-connected versus poorly connected agents is exacerbated in the presence of random *link failures*, where any message sent by an agent can fail to reach its recipient with a failure probability $1 - p$ (drawn i.i.d. for each message).

Indeed, it is natural to expect the group regret to decrease with decreasing link failure probability, i.e., increasing communication probability $p$. However, what we observe experimentally (Section 7) is that this holds only for graphs $G$ that are *regular* (i.e., each agent has the same degree), or close to regular. When $G$ is irregular, as we increase $p$ from 0 to 1, the group performance oscillates. While, in some cases, the improved performance in the well-connected agents can outweigh the degradation observed in the weakly-connected agents (leading to lower *group* regret), it is prudent to consider an approach that mitigates this disparity by regulating information flow in the network.

**Information Regulation in Cooperative Bandits**. Our approach to regulate information is straightforward: we direct each agent $i$ to discard any incoming message with an agent-specific probability $1 - p_i$, while always utilizing its own observations. For specific values of $p_i$, we can obtain various weighted combinations of internal versus group observations. Our first algorithm RCL-LF (Link Failures) is built on this regulation strategy, coupled with UCB1 exploration using all selected observations for each arm. Essentially, each agent runs UCB1 using the cumulative set of observations it has received from its network. After pulling an arm, it broadcasts its pulled arm and reward through the network, but incorporates each incoming message *only* with a probability $p_i$. Pseudo code for the algorithm is given in the appendix. We first present a regret bound for RCL-LF when run with the *instantaneous reward-sharing* protocol.

**Theorem 1** (RCL-LF Regret with instantaneous reward-sharing)**.** RCL-LF *running with the instantaneous reward-sharing protocol (Figure 1, $\gamma = 1$) obtains cumulative group regret of*

$$\mathrm{Reg}_G(T) \leq g(\xi, \sigma) \left( \sum_{i=1}^{N} (1 - p_i \cdot p) + \sum_{\mathcal{C} \in \mathfrak{C}} (\max_{i \leq \mathcal{C}} p_i) \cdot p \right) \left( \sum_{k>1} \frac{\log T}{\Delta_k} \right) + f(5N, G)$$

*where $\mathfrak{C}$ is a non-overlapping clique covering of $G$.*

*Proof sketch.* We follow an approach similar to the analysis of UCB1 by [3] with several key modifications. First, we partition the communication graph $G$ into a set of non-overlapping cliques and then analyze the regret of each clique. The group regret can be obtained by taking the summation of the regret over each clique. Two major technical challenges in proving the regret bound for RCL-LF are (a) deriving a tail probability bound for probabilistic communication, and (b) bounding the regret accumulated by agents by losing information due to communication failures and message discarding. We overcome the first challenge by noticing that communication is independent of the decision making process thus $\mathbb{E} \left( \exp \left( \lambda \sum_{\tau=1}^{t} X_\tau^i \mathbf{1}\{A_\tau^i = k\} - \mu_k N_k^i(t) - \frac{\lambda^2 \sigma_k^2}{2} N_k^i(t) \right) \right) \leq 1$ holds under probabilistic communication. We obtain the tail bound by combining this result with the Markov inequality and optimizing over $\lambda$ using a peeling type argument. We address the second challenge by proving that the number of times agents do not share information about any suboptimal arm $k$ can be bounded by a term that increases logarithmically with time and scales with number of agents, $G$, and communication probabilities, as $\sum_{i=1}^{N} (1 - p_i \cdot p) + \sum_{\mathcal{C} \in \mathfrak{C}} (\max_{i \leq \mathcal{C}} p_i) \cdot p$. $\square$

**Remark 1** (Regret bound optimality)**.** Under perfect communication ($p = 1$) and no message discarding, i.e., $p_i = p = 1, \forall i \in [N]$ the dominant term in our regret bound scales with $\bar{\chi}(G)$, obtaining identical performance to deterministic communication over $G$ [11]. Alternatively, when $p_i = p = 0$, there is no communication, and hence, the regret bound is $\mathcal{O}(N \log T)$. Theorem 1 quantifies the benefit of communication in reducing the group regret under probabilistic link failure and when agents incorporate observations with an agent-specific probability. Note that $\sum_{i=1}^{N} (1 - p_i \cdot p) + \sum_{\mathcal{C} \in \mathfrak{C}} (\max_{i \leq \mathcal{C}} p_i) \cdot p = N - p \cdot \left( \sum_{i=1}^{N} p_i - \sum_{\mathcal{C} \in \mathfrak{C}} (\max_{i \leq \mathcal{C}} p_i) \right)$. Since the clique covering is non-overlapping, the results show that agents obtain improved group performance for any communication probability $p > 0$ for any nontrivial graph as compared to the case with no communication in which each agent learns on its own.

**Remark 2** (Controlling information disparity)**.** In order to regulate the information disparity across the network we set $p_i = \frac{d_{\min}(G)}{d_i(G)}$. Thus, the agent(s) with minimum degree $d_{\min}$ incorporate each message they receive with probability 1 and we have that the *expected* number of messages for each agent is the same, i.e., $T \cdot d_{\min}(G)$. Therefore, every agent receives the same amount of information (in expectation), providing a large performance improvement for irregular graphs (see Section 7).

**Message-Passing**. Under this communication protocol each agent $i$ communicates with neighbors at distance at most $\gamma$, where each hop adds a 1-step delay. Our algorithm RCL-CF obtains a similar regret bound in this setting as well, when all agents use the same UCB1 exploration strategy (Figure 2).

**Theorem 2** (RCL-LF Regret with message-passing). *Let $\mathcal{C}$ be a minimal clique covering of $G_\gamma$. For any $\mathcal{C} \in \mathcal{C}$ and $i, j \in \mathcal{C}$ let $\gamma_i = \max_{j \in \mathcal{C}} d(i,j)$ be the maximum distance (in graph $G$) between agents $i$ and $j$. RCL-LF running with the message-passing protocol (Figure 1) with delay parameter $\gamma$ obtains cumulative group regret of*

$$\mathrm{Reg}_G(T) \le g(\xi, \sigma) \left( \sum_{i=1}^{N} (1 - p_i \cdot p^{\gamma_i}) + \bar{\chi}(G_\gamma) \cdot (\max_{i \le N} p_i \cdot p^{\gamma_i}) \right) \left( \sum_{k>1} \frac{\log T}{\Delta_k} \right) + f((\gamma + 4)N, G_\gamma).$$

*Proof sketch.* We partition the graph $G_\gamma$ into non-overlapping cliques, analyze the regret of each clique and take the summation of regrets over cliques to obtain group regret. In addition to the challenges encountered in Theorem 1 here we are required to account for having different probabilities of failures for messages due to having multiple paths of different length between agents and to account for the delay incurred by each hop when passing messages. We overcome the first challenge by noting that agent $i$ receives each message with at least probability $p^{\gamma_i}$. We overcome the second challenge by identifying that regret incurred by delays can be upper bounded using $\left( \sum_{i=1}^{N} \gamma_i - N \right) \sum_{k>1} \Delta_k$. □

**Remark 3.** Finding an optimal observation probability $\{p_i\}_{i=1}^{N}$ for RCL-LF with message-passing is difficult due to the delays added by each hop when forwarding messages. If messages are forwarded without a delay, optimal performance can be obtained by using $p_i = \frac{d_{\min}(G_\gamma)}{d_i(G_\gamma)}$. For dense $G_\gamma$, the above choice of observation probability provides *near*-optimal performance. When $\gamma = d_\star(G)$ we have that $G_\gamma$ is a complete graph, $p_i = \frac{d_{\min}(G_\gamma)}{d_i(G_\gamma)} = 1$, and agents do not discard any message. However, when $\gamma < d_\star(G)$, the graph $G_\gamma$ is not complete. Therefore agents receive different amounts of information which are approximately proportional to the degree distribution of $G_\gamma$. As explained earlier this information disparity leads to a performance disparity among agents. As a result group performance decreases. In this case we design the algorithm such that each agent $i$ discards messages with $1 - p_i$ where $p_i = \frac{d_{\min}(G_\gamma)}{d_i(G_\gamma)}$. This regulates the information flow mitigating the bias introduced by information disparity. As a result the group obtains near-optimal performance.

## 4 Instantaneous Reward-sharing Under Stochastic Delays

Next, we consider a communication protocol, where any message is received after an arbitrary (but bounded) stochastic delay. We assume for simplicity that each message is sent only once in the network (and not forwarded multiple times as in *message-passing*), and leave the message-passing setting as future work. We assume, furthermore that the delays are identically and independently drawn from a bounded distribution with expectation $\mathbb{E}[\tau]$ (similar to prior work, e.g., Joulani et al. [18], Vernade et al. [33]). For this setting, we demonstrate that cooperative UCB1, along with incorporating all messages as soon as they are available, provides efficient performance, both empirically and theoretically. We denote this algorithm as RCL-SD (Stochastic Delays), and demonstrate that this approach incurs only an extra $\mathcal{O}(\sqrt{N \log T} + \log T)$ overhead compared to perfect communication.

**Theorem 3** (RCL-SD Regret). *Let $D_{total} = N \cdot \mathbb{E}[\tau] + 2\log T + 2\sqrt{N \cdot \mathbb{E}[\tau] \log T}$ denote an upper bound on the total number of outstanding messages. RCL-SD obtains, with probability at least $1 - \frac{1}{T}$, cumulative group regret of*

$$\mathrm{Reg}_G(T) \le g(\xi, \sigma) \cdot \bar{\chi}(G) \cdot \left( \sum_{k>1} \frac{\log T}{\Delta_k} \right) + D_{total} \cdot \left( \sum_{k>1} \Delta_k \right) + f(5N, G).$$

*Proof sketch.* We first demonstrate that the additional group regret due to stochastic delays can be bounded by the maximum number of cumulative outstanding messages over all agents at any given time step. Then we apply a result similar to Lemma 2 of [18] to bound the total number of outstanding messages using the cumulative expected delay $N \cdot \mathbb{E}[\tau]$, giving the result. □

**Remark 4.** The $D_{total}$ term is a succinct upper bound on the maximum number of cumulative outstanding messages over all agents, and when the expected delay $\mathbb{E}[\tau] = o(1)$, we see that the contribution of $D_{total}$ is $\mathcal{O}(\sqrt{N \log T} + \log T)$. We conjecture that this cannot be improved without restricting communication, as each agent will send $T$ messages in total. The result obtained by Joulani et al. [18] has a similar dependence for a single agent.

**Algorithm 1:** `RCL-RC`: Cooperative Hybrid Arm Elimination

---

**Parameters:** Confidence $\delta \in (0,1)$, horizon $T$, graph $G$ with exploration set $\mathcal{I} \subseteq \mathcal{V}$. Initialize $T_i(0) = K, \forall i \in \mathcal{I} \lambda = 1024 \log \left( \frac{8K\psi(G_\gamma)}{\delta} \log_2 T \right)$ and $\Delta_k^i(0) = 1, \forall\, k \in [K]$ and $i \in \mathcal{I}$.

**for** each subgraph $\mathcal{N}_i^+(G_\gamma)$ where $i \in \mathcal{I}$ **do**
 **for** $t = 1, ..., K$, each agent $j \in \mathcal{N}_i^+(G_\gamma)$ **do**
  | Play arm $K$ and get reward $r_j(t)$.
 **end**
 **for** epoch $m_i = 1, 2, ...,$ **do**
  Set $n_k^i(m_i) = \lambda(\Delta_k^i(m_i - 1))^{-2} \forall k \in [K]$.
  $N_i(m_i) = \sum_k n_k^i(m_i)$ and $T_i(m_i) = T_i(m_i) + N_i(m_i) + 2\gamma$.
  **for** agent $j \in \mathcal{N}_i^+(G_\gamma)$ **do**
   **for** $t = T_i(m_i - 1)$ *to* $s = T_i(m_i - 1) + 2\gamma$ **do**
    **if** $j \neq i$ **then**
     **if** $t \leq K + d(i,j)$ **then**
      | Pull random arm.
     **end**
     **else**
      | Pull $A_j(t) = A_i(t - d(i,j))$ and get reward $r_j(t)$.
     **end**
    **end**
    **else**
     | Pull $A_j(t) = \texttt{UCB1}(t)$
    **end**
   **end**
   **for** $t = T_i(m_i - 1) + 2\gamma$ *to* $T_i(m_i)$ **do**
    **if** $j \neq i$ **then**
     | Pull $A_j(t) = A_i(t - d(i,j))$ and get reward $r_j(t)$.
    **end**
    **else**
     | Pull an arm $A_i(t) = k \in [K]$ with probability $n_k^i(m_i)/N_k(m_i)$.
    **end**
   **end**
  **end**
 **end**
**end**

---

## 5 Hybrid Arm Elimination for Adversarial Reward Corruptions

In this section, we assume that any reward when transmitted can be corrupted by a maximum value of $\epsilon$, i.e., $\max_{t,n} |r_n(t) - \tilde{r}_n(t)| \leq \epsilon$ where $\tilde{r}_n(t)$ denotes the transmitted reward. Furthermore, we assume that the corruptions can be *adaptive*, i.e., can depend on the prior actions and rewards of each agent. This model includes natural settings, where messages can be corrupted during transmission, as well as *byzantine* communication [12]. If $\epsilon$ were known, we could then extend algorithms for misspecified bandits [14] to create a robust estimator and a subsequent `UCB1`-like algorithm that obtains a regret of $\mathcal{O}(\bar{\chi}(G_\gamma)K(\frac{\log T}{\Delta}) + TNK\epsilon)$. However, this approach has two issues. First, $\epsilon$ is typically not known, and the dependence on $G_\gamma$ can be improved as well. We present an arm-elimination algorithm called `RCL-AC` (Adversarial Corruptions) that provides better guarantees on regret, without knowledge of $\epsilon$ in Algorithm 1.

The central motif in `RCL-AC`'s design is to eliminate bad arms by an epoch-based exploration, an idea that has been successful in the past for adversarially-corrupted stochastic bandits [25, 15]. The challenge, however, in a message-passing decentralized setting is two-fold. First, agents have different amounts of information based on their position in the network, and hence badly positioned agents in $G$ may be exploring for much larger periods. Secondly, communication between agents is delayed, and hence any agent naively incorporating stale observations may incur a heavy bias from delays. To ameliorate the first issue, we partition the group of agents into two sets - *exploring agents* ($\mathcal{I}$) and

*imitating agents* ($\mathcal{V} \setminus \mathcal{I}$). The idea is to only allow well-positioned agents in $\mathcal{I}$ to direct the exploration strategy for their neighboring agents, and the rest simply imitate their exploration strategy. We select $\mathcal{I}$ as a minimal dominating set of $G_\gamma$, hence $|\mathcal{I}| = \psi(G_\gamma)$. Furthermore, since $\mathcal{V} \setminus \mathcal{I}$ is a vertex cover, this ensures that each *imitating* agent is connected (at distance at most $\gamma$) to at least one agent in $\mathcal{I}$. Next, observe that there are two sources of delay: first, any *imitating* agent must wait at most $\gamma$ trials to observe the latest action from its corresponding *exploring* agent, and second, each *exploring* agent must wait an additional $\gamma$ trials for the feedback from all of its imitating agents. We propose that each *exploring* agent run UCB1 for $2\gamma$ rounds after each epoch of arm elimination, using *only* local pulls. This prevents a large bias due to these delays, at a small cost of $\mathcal{O}(\log \log T)$ suboptimal pulls.

**Theorem 4** (RCL-RC Regret)**.** RCL-RC *obtains, with probability at least $1 - \delta$, group regret of*

$$\mathsf{Reg}_G(T) = \mathcal{O}\left( KTN\gamma\epsilon + \psi(G_\gamma) \cdot \sum_{k>1} \frac{\log T}{\Delta_k} \log\left( \frac{K\psi(G_\gamma)\log T}{\delta} \right) + N\sum_{k>1} \Delta_k + \sum_{k>1} \frac{N\log(\gamma \log T)}{\Delta_k} \right).$$

*Proof sketch.* Since the dominating set covers $\mathcal{V}$, we can decompose the group regret into the cumulative regret of the subgraphs corresponding to each agent in $\psi(G_\gamma)$. For each subgraph, we can consider the cumulative regret incurred when the *exploring agent* follows UCB1 versus arm elimination. We have that arm elimination occurs for $\log T$ epochs, and since UCB1 runs for $2\gamma$ rounds between succesive epochs, we have that in any subgraph of size $n$, the cumulative regret from UCB1 rounds is of $\mathcal{O}(nK\log(\gamma \log T))$. For arm elimination, we can bound the subgraph regret using a modification of the approach in Gupta et al. [15]: the difference in our approach is to construct a multi-agent filtration for arbitrary (reward-dependent) corruptions from message-passing, and then applying Freedman's bound on the resulting martingale sequence. Subsequently, the regret in each epoch is bounded in a manner similar to Gupta et al. [15], and finally applying a union bound. $\square$

**Remark 5** (Regret Optimality)**.** Theorem 4 demonstrates a trade-off between communication density and the adversarial error, as seen by the first two terms in the regret bound. The first term ($KTN\gamma\epsilon$) is a bound on the cumulative error introduced due to message-passing, which is increasing in $\gamma$, whereas the second term denotes the logarithmic regret due to exploration, where $\psi(G_\gamma)$ decreases as $\gamma$ increases: for $\gamma = d_\star(G), \psi(G_\gamma) = 1$, matching the lower bound in Dubey and Pentland [11]. This too is expected, as fewer exploring agents are needed with a higher communication budget. Furthermore, we conjecture that the first term is optimal (in terms of $T$, up to graphical constants): a linear lower bound has been demonstrated for the single-agent setting in Lykouris et al. [25].

**Remark 6** (Computational complexity)**.** While the dominating set problem is known to be NP-complete [19], the problem admits a polynomial-time approximation scheme (PTAS) [9] for certain graphs, for which our bounds hold exactly. However, RCL-RC can work on any dominating set of size $n$, and suffer regret of $\widetilde{\mathcal{O}}(KTN\gamma\epsilon + n\sum_{k>1} \frac{\log T}{\Delta_k})$[1].

## 6 An Algorithm for Perfect Communication and Lower Bounds

For perfect communication, we present Delayed MP-UCB, a simple improvement to UCB1 with message-passing where each agent $i$ only incorporates messages originated prior to $\bar{\gamma} \leq \gamma$ time steps, reducing disparity in information across agents.

**Theorem 5** (Delayed MP-UCB Regret)**.** Delayed(MP)-UCB *obtains cumulative group regret of*

$$\mathsf{Reg}_G(T) \leq g(\xi, \sigma)\bar{\chi}(G_\gamma) \left( \sum_{k>1} \frac{\log T}{\Delta_k} \right) + (N - \bar{\chi}(G_\gamma)(\gamma - 1)\sum_{k>1} \Delta_k + f(5N, G_\gamma) + h(G_\gamma, \bar{\gamma})$$

*where* $h(G_\gamma, \bar{\gamma}) = \left( (N - \bar{\chi}(G_\gamma)\bar{\gamma} + \sum_{t > \bar{\gamma}}^T \frac{\log\left(1 - \frac{d_i(G_\gamma)\bar{\gamma}}{(d_i(G_\gamma)+1)t}\right)}{\log 1.3} \frac{1}{t^{(\xi+1)\left(1 - \frac{0.09}{16}\right)}} \right) \sum_{k>1} \Delta_k.$

*Proof sketch.* Following a similar approach to the proof of Theorem 2 we partition the graph $G_\gamma$ into a set of non-overlapping cliques, analyze the regret of each clique via a UCB1 type analysis and take the summation of regret over cliques. However, using less information (due to delayed information usage) in estimates leads to a large confidence bound $C_k^i(t)$ and this reduces the contribution to the regret from tail probabilities. Note that $\log\left(1 - \frac{d_i(G_\gamma)\bar{\gamma}}{(d_i(G_\gamma)+1)t}\right)$ is negative $\forall t > \bar{\gamma}$, and hence lower regret achieved due to low tail probabilities is given by the second term of $h(G_\gamma, \bar{\gamma})$. $\square$

---

[1]The $\widetilde{\mathcal{O}}$ notation ignores absolute constants and $\log\log(\cdot)$ factors in $T$.

**Remark 7.** Incorporating only the messages originated before $\bar{\gamma}$ time steps is similar to communicating over $G_{\bar{\gamma}}$ after a delay of $\bar{\gamma}$ time steps. When $G$ is connected and $\bar{\gamma} = \gamma = d_*$ this is similar to communicating over a complete graph with a delay of $d_*$. Thus Delayed MP-UCB mitigates the disparity in information used by each agent, leading to improved group performance.

**Lower Bounds.** Without strict assumptions, a lower bound of $\mathcal{O}\left(\sum_{k>1} \log T / \Delta_k\right)$ has been demonstrated both for $\gamma = 1$ (instantaneous reward-sharing, Kolla et al. [20]) and $\gamma > 1$ (message-passing, Dubey and Pentland [11]), which both suggest that a speedup of $\frac{1}{N}$ is potentially achievable. For a more restrictive class of *individually consistent* and *non-altruistic* policies (i.e., that do not contradict their local feedback), a tighter lower bound of $\mathcal{O}\left(\alpha(G_2) \sum_{k>1} \log T / \Delta_k\right)$ can be demonstrated for reward-sharing [20], and consequently $\mathcal{O}\left(\alpha(G_{\gamma+1}) \sum_{k>1} \log T / \Delta_k\right)$ for message-passing. To supplement these results, we present a lower bound to characterize the minimax optimal rates for the problem. We present first an assumption on multi-agent policies.

**Assumption 1** (Agnostic decentralized policies). *A set of $N$ policies $\pi_1, ..., \pi_N$ are termed agnostic decentralized policies, if for every pair $(i, j)$ of agents that communicate in $G$ and each $t \in [T]$, $\pi_i(t)$ is independent of $\{\pi_j(\tau)\}_{\tau=1}^{t-d(i,j)}$ conditioned on the rewards $\{(A_j(\tau), X_j(\tau))\}_{\tau=1}^{t-d(i,j)}$.*

**Theorem 6** (Minimax Rate). *For any policy $\mathcal{A}$, there exists a $K$-armed environment over $N$ agents with $\Delta_k \leq 1$ for any connected graph $G$ and $\gamma \geq 1$ such that, for some absolute constant $c$,*

$$\mathsf{Reg}_G(\mathcal{A}, T) \geqslant c\sqrt{KN(T + \widetilde{d}(G))}.$$

*Furthermore, if $\mathcal{A}$ is an agnostic decentralized policy, there exists a $K$-armed environment over $N$ agents with $\Delta_k \leq 1$ for any connected graph $G$ and $\gamma \geq 1$ such that, for some absolute constant $c'$,*

$$\mathsf{Reg}_G(\mathcal{A}, T) \geqslant c'\sqrt{\alpha^\star(G_\gamma)KNT}.$$

*Here $\tilde{d}(G) = \sum_{i=1}^{d^\star(G)} \bar{d}_{=i} \cdot i$ denotes the average delay incurred by message-passing across the network $G$, and $\alpha^\star(G_\gamma) = \frac{N}{1+\bar{d}_\gamma}$ is Turan's lower bound [32] on $\alpha(G_\gamma)$.*

**Remark 8** (Tightness of lower bound). The first minimax bound does not make any assumptions on the policy $\mathcal{A}$, and hence we only see an additive dependence of the average delay incurred by communication over $G$. This dependence generalizes the minimax rate for delayed multi-armed bandits [30] to graphical feedback. For the latter bound, observe that a variety of cooperative extensions of single-agent bandit algorithms [20, 11, 7] obey this assumption, where the decision-making for any agent is independent of any other agent, conditioned on the observed rewards. In this setting, agents merely treat messages as additional pulls to construct stronger estimators, and do not strategize collectively. This bound is exact (up to constants) for a variety of communication graphs $G$. For instance, for linear and circular graphs, $\frac{\alpha^\star(G_\gamma)}{\alpha(G_\gamma)} = o(1)$, and for $d$-regular graphs, $\alpha^\star(G_\gamma) = \alpha(G_\gamma)$ [32].

# 7 Experimental Results

We consider the 10-armed bandit with rewards drawn from Gaussian distributions with $\sigma_k = 1$ for each arm, such that $\mu_1 = 1$ and $\mu_k = 0.5$ for $k \neq 1$, and the number of agents $N = 50$, where we repeat each experiment 100 times with $G$ selected randomly from different families of random graphs. The bottom row of Figure 3 corresponds to Erdos-Renyi graphs with $p = 0.7$. The top row of Figure 3 (a), (c) and (d) corresponds to multi-star graphs and (b) and (e) to random tree graphs. We set $\xi = 1.1$ and $\gamma = \max\{3, d_\star(G)/2\}$.

**Stochastic Link Failure.** Figure 3(a) and Figure 3(b) summarize performance of `RCL(RS)-LF` and `RCL(MP)-LF`, comparing it with the corresponding reward-sharing and message-passing UCB-like algorithms in which $p_i = 1, \forall i \in [N]$, for different $p$ values. The group regret is given at $T = 500$. The results validate our claim that probabilistic message discarding improves performance for irregular graphs and provides competitive performance for *near*-regular graphs.

**Stochastic Delays.** We compare performance of `RCL-SD` with `UCB1`. We draw delays from a bounded distribution with $\mathbb{E}[\tau] = 10$ and $\tau_{\max} = 50$. The results are summarized in Figure 3(c).

**Adversarial Communication.** We compute the (approximate) dominating set using the algorithm provided in `networkx` for each connected component in $G_\gamma$. We draw corruptions uniformly from the

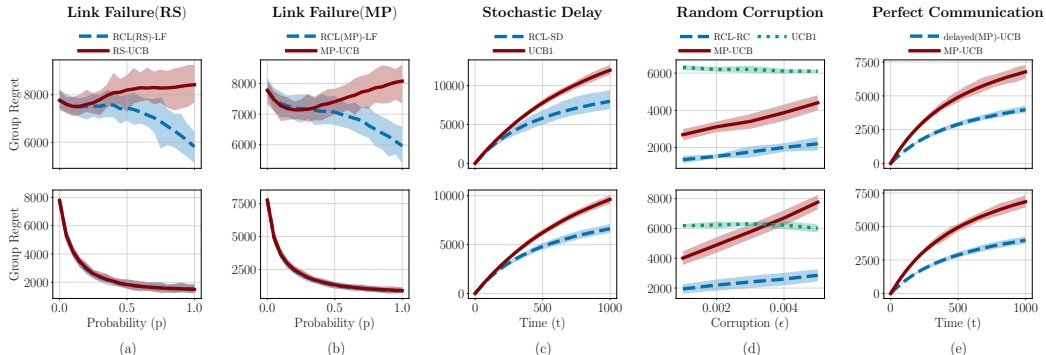

Figure 3: Experimental results for various imperfect communication settings.

range $[0, \epsilon]$ for each message, where $\epsilon$ is increased from $10^{-3}$ to $10^{-2}$. The group regret at $T = 500$ as a function of $\epsilon$ is shown in Figure 3(d) and compared against individual `UCB1` and cooperative UCB with message-passing (`MP-UCB`), which incur larger regret increasing linearly with $\epsilon$.

**Perfect Communication**. We compare the regret curve for $T = 1000$ for our `Delayed(MP)-UCB` against regular `MP-UCB` in Figure 3(e). We use $\bar{\gamma} = 2$. It is evident that delayed incorporation of messages markedly improves performance across both networks.

## 8   Conclusions

In this paper, we studied the cooperative bandit problem in three different imperfect communication settings. For each setting, we proposed algorithms with competitive empirical performance and provided theoretical guarantees on the incurred regret. Further, we provided an algorithm for perfect communication that comfortably outperforms existing baseline approaches. We additionally provided a tighter network-dependent minimax lower bound for the cooperative bandit problem. We believe that our contributions can be of immediate utility in applications. Moreover, future inquiry can be pursued in several different directions, including multi-agent reinforcement learning and contextual bandit learning.

**Ethical Considerations**. Our work is primarily theoretical, and we do not foresee any negative societal consequences arising specifically from our contributions in this paper.

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
