# A   Proof of Theorem 1

We consider the case where each message fails with probability $1 - p$ and each agent $i$ uses the messages it receives from its neighbors with probability $p_i$. This is equivalent to each agent $i$ receiving messages from its neighbors with probability $p_i p$. Let $\mathbf{1}\{(i, j) \in E_t\}$ be the indicator random variable that takes value 1 if agent $i$ receives reward value and arm id from agent $j$ at time $t$ and 0 otherwise.

We start by proving some useful lemmas.

**Lemma 1. (Restatement of results from [3])** *Let* $\eta_k = \left( \frac{8(\xi+1)\sigma^2}{\Delta_k^2} \right) \log T$. *For any suboptimal arm* $k$ *and* $\forall i, t$ *we have*

$$\mathsf{P}\left( A_i(t+1) = k, N_k^i(t) > \eta_k \right) \leq \mathsf{P}\left( \widehat{\mu}_1^i(t) \leq \mu_1 - C_1^i(t) \right) + \mathsf{P}\left( \widehat{\mu}_k^i(t) \geq \mu_k + C_k^i(t) \right)$$

*Proof.* Let $Q_k^i(t) = \widehat{\mu}_k^i(t) + C_k^i(t)$. Note that for any $k > 1$ we have

$$\{A_i(t+1) = k\} \subset \{Q_k^i(t) \geq Q_1^i(t)\}$$
$$\subset \left\{ \{\mu_1 < \mu_k + 2C_k^i(t)\} \cup \{\widehat{\mu}_1^i(t) \leq \mu_1 - C_1^i(t)\} \cup \{\widehat{\mu}_k^i(t) \geq \mu_k + C_k^i(t)\} \right\}.$$

Let $\eta_k = \left( \frac{8(\xi+1)\sigma^2}{\Delta_k^2} \right) \log T$. Since $N_k^i(t) > \eta_k$ the event $\{\mu_1 < \mu_k + 2C_k^i(t)\}$ does not occur. Thus we have

$$\mathsf{P}\left( A_i(t+1) = k, N_k^i(t) > \eta_k \right) \leq \mathsf{P}\left( \widehat{\mu}_1^i(t) \leq \mu_1 - C_1^i(t) \right) + \mathsf{P}\left( \widehat{\mu}_k^i(t) \geq \mu_k + C_k^i(t) \right)$$

This concludes the proof of Lemma 1. $\qquad \square$

**Lemma 2.** *Let* $\bar{\chi}(G)$ *is the clique covering number of graph* $G$. *Let* $\eta_k = \left( \frac{8(\xi+1)\sigma_k^2}{\Delta_k^2} \right) \log T$. *Then we have*

$$\sum_{i=1}^{N} \mathbb{E}[n_k^i(T)] \leq \left( \sum_{i=1}^{N} (1 - p_i p) + \bar{\chi}(G) p_{\max} p \right) \eta_k + 2N \tag{1}$$

$$+ \sum_{i=1}^{N} \sum_{t=1}^{T-1} \left[ \mathsf{P}\left( \widehat{\mu}_1^i(t) \leq \mu_1 - C_1^i(t) \right) + \mathsf{P}\left( \widehat{\mu}_k^i(t) \geq \mu_k + C_k^i(t) \right) \right] \tag{2}$$

*Proof.* Let $\mathcal{C}$ be a non overlapping clique covering of $G$. Note that for each suboptimal arm $k > 1$ we have

$$\sum_{i=1}^{N} \mathbb{E}[n_k^i(T)] = \sum_{i=1}^{N} \sum_{t=1}^{T} \mathsf{P}\left( A_i(t) = k \right) = \sum_{C \in \mathcal{C}} \sum_{i \in C} \sum_{t=1}^{T} \mathsf{P}\left( A_i(t) = k \right). \tag{3}$$

Let $\tau_{k,\mathcal{C}}$ denote the maximum time step when the total number of times arm $k$ has been played by all the agents in clique $\mathcal{C}$ is at most $\eta_k + |\mathcal{C}|$ times. This can be stated as $\tau_{k,\mathcal{C}} := \max\{t \in [T] : \sum_{i \in \mathcal{C}} n_k^i(t) \leq \eta_k + |\mathcal{C}|\}$. Then, we have that $\eta_k < \sum_{i \in \mathcal{C}} n_k^i(\tau_{k,\mathcal{C}}) \leq \eta_k + |\mathcal{C}|$.

For each agent $i \in \mathcal{C}$ let

$$\bar{N}_k^i(t) := \sum_{j \in \mathcal{C}} \sum_{\tau=1}^{t} \mathbf{1}\{A_j(\tau) = k\} \mathbf{1}\{(i, j) \in E_\tau\},$$

denote the sum of the total number of times agent $i$ pulled arm $k$ and the total number of observations it received from agents in its clique about arm $k$ until time $t$. Define $\bar{\tau}_{k,\mathcal{C}}^i := \max\{t \in [T] : \bar{N}_k^i(t) \leq \eta_k\}$. Then we have that $\eta_k - |\mathcal{C}| < \bar{N}_k^i(\bar{\tau}_{k,\mathcal{C}}^i) \leq \eta_k$.

Note that $N_k^i(t) \geq \bar{N}_k^i(t), \forall t$, hence for all $i \in \mathcal{C}$ we have $N_k^i(t) > \eta_k, \forall t > \bar{\tau}_{k,\mathcal{C}}^i$. Here we consider that $\bar{\tau}_{k,\mathcal{C}}^{(i)} \geq \tau_{k,\mathcal{C}}, \forall i$. From regret results it follows that regret for this case is greater than the regret for the case where $\bar{\tau}_{k,\mathcal{C}}^i < \tau_{k,\mathcal{C}}$ for some (or all) $i$.

We analyse the expected number of times agents pull suboptimal arm $k$ as follows,

$$\sum_{\mathcal{C}\in\mathfrak{C}}\sum_{i\in\mathcal{C}}\sum_{t=1}^{T}\mathbf{1}\{A_i(t)=k\} \tag{4}$$

$$=\sum_{\mathcal{C}\in\mathfrak{C}}\sum_{i\in\mathcal{C}}\sum_{t=1}^{\tau_{k,\mathcal{C}}}\mathbf{1}\{A_i(t)=k\}+\sum_{\mathcal{C}\in\mathfrak{C}}\sum_{i\in\mathcal{C}}\sum_{t>\tau_{k,\mathcal{C}}}^{\bar{\tau}_{k,\mathcal{C}}^{i}}\mathbf{1}\{A_i(t)=k\}+\sum_{\mathcal{C}\in\mathfrak{C}}\sum_{i\in\mathcal{C}}\sum_{t>\bar{\tau}_{k,\mathcal{C}}^{i}}^{T}\mathbf{1}\{A_i(t)=k\} \tag{5}$$

$$\leq\sum_{\mathcal{C}\in\mathfrak{C}}(\eta_k+|\mathcal{C}|)+\sum_{\mathcal{C}\in\mathfrak{C}}\sum_{i\in\mathcal{C}}\sum_{t>\tau_{k,\mathcal{C}}}^{\bar{\tau}_{k,\mathcal{C}}^{i}}\mathbf{1}\{A_i(t)=k\}+|\mathcal{C}| \tag{6}$$

$$+\sum_{\mathcal{C}\in\mathfrak{C}}\sum_{i\in\mathcal{C}}\sum_{t>\bar{\tau}_{k,\mathcal{C}}^{i}}^{T-1}\mathbf{1}\{A_i(t+1)=k\}\mathbf{1}\{N_k^i(t)>\eta_k\}. \tag{7}$$

Taking expectation we have

$$\sum_{\mathcal{C}\in\mathfrak{C}}\sum_{i\in\mathcal{C}}\sum_{t=1}^{T}\mathsf{P}\left(A_i(t)=k\right)\leq\sum_{\mathcal{C}\in\mathfrak{C}}(\eta_k+2|\mathcal{C}|) \tag{8}$$

$$+\sum_{\mathcal{C}\in\mathfrak{C}}\sum_{i\in\mathcal{C}}\sum_{t>\tau_{k,\mathcal{C}}}^{\bar{\tau}_{k,\mathcal{C}}^{i}}\mathsf{P}\left(A_i(t)=k\right)+\sum_{\mathcal{C}\in\mathfrak{C}}\sum_{i\in\mathcal{C}}\sum_{t>\bar{\tau}_{k,\mathcal{C}}^{i}}^{T-1}\mathsf{P}\left(A_i(t+1)=k,N_k^i(t)>\eta_k\right). \tag{9}$$

Note that we have

$$\sum_{i\in\mathcal{C}}\sum_{t>\tau_{k,\mathcal{C}}}^{\bar{\tau}_{k,\mathcal{C}}^{i}}\mathbf{1}\{A_i(t)=k\} \tag{10}$$

$$=\sum_{i\in\mathcal{C}}\bar{N}_k^i(\bar{\tau}_{k,\mathcal{C}}^{i})-\sum_{i\in\mathcal{C}}\sum_{t=1}^{\tau_{k,\mathcal{C}}}\mathbf{1}\{A_i(t)=k\}-\sum_{i\in\mathcal{C}}\sum_{j\neq i,j\in\mathcal{C}}\sum_{t=1}^{\bar{\tau}_{k,\mathcal{C}}^{i}}\mathbf{1}\{A_j(t)=k\}\mathbf{1}\{(i,j)\in E_t\} \tag{11}$$

$$=\sum_{i\in\mathcal{C}}\bar{N}_k^i(\bar{\tau}_{k,\mathcal{C}}^{i})-\sum_{i\in\mathcal{C}}n_k^i(\tau_{k,\mathcal{C}})-\sum_{i\in\mathcal{C}}\sum_{j\neq i,j\in\mathcal{C}}\sum_{t=1}^{\bar{\tau}_{k,\mathcal{C}}^{i}}\mathbf{1}\{A_j(t)=k\}\mathbf{1}\{(i,j)\in E_t\} \tag{12}$$

$$\leq|\mathcal{C}|\eta_k-\eta_k-\sum_{i\in\mathcal{C}}\sum_{j\neq i,j\in\mathcal{C}}\sum_{t=1}^{\bar{\tau}_{k,\mathcal{C}}^{i}}\mathbf{1}\{A_j(t)=k\}\mathbf{1}\{(i,j)\in E_t\} \tag{13}$$

$$\leq|\mathcal{C}|\eta_k-\eta_k-\sum_{i\in\mathcal{C}}\sum_{j\neq i,j\in\mathcal{C}}\sum_{t=1}^{\tau_{k,\mathcal{C}}}\mathbf{1}\{A_j(t)=k\}\mathbf{1}\{(i,j)\in E_t\}. \tag{14}$$

Taking the expectation

$$\sum_{i\in\mathcal{C}}\sum_{t>\tau_{k,\mathcal{C}}}^{\bar{\tau}_{k,\mathcal{C}}^{i}}\mathsf{P}\left(A_i(t)=k\right)\leq|\mathcal{C}|\eta_k-\eta_k-\sum_{i\in\mathcal{C}}p_ip\sum_{j\neq i,j\in\mathcal{C}}\sum_{t=1}^{\tau_{k,\mathcal{C}}}\mathsf{P}\left(A_j(t)=k\right) \tag{15}$$

$$=|\mathcal{C}|\eta_k-\eta_k-\sum_{i\in\mathcal{C}}p_ip\sum_{j\neq i,j\in\mathcal{C}}\mathbb{E}(n_k^j(\tau_{k,\mathcal{C}})) \tag{16}$$

$$=|\mathcal{C}|\eta_k-\eta_k-\left(\sum_{i\in\mathcal{C}}p_ip\right)\left(\sum_{i\in\mathcal{C}}\mathbb{E}(n_k^i(\tau_{k,\mathcal{C}}))\right)+\sum_{i\in\mathcal{C}}p_ip\mathbb{E}(n_k^i(\tau_{k,\mathcal{C}})) \tag{17}$$

$$\leq|\mathcal{C}|\eta_k-\eta_k-p\left(\sum_{j\in\mathcal{C}}p_j-p_{\max}\right)\mathbb{E}\left(\sum_{i\in\mathcal{C}}n_k^i(\tau_{k,\mathcal{C}})\right) \tag{18}$$

$$\le |\mathcal{C}|\eta_k - \eta_k - p\left(\sum_{j\in\mathcal{C}} p_j - p_{\max}\right)\eta_k \tag{19}$$

$$= \left(|\mathcal{C}| - 1 - p\left(\sum_{j\in\mathcal{C}} p_j - p_{\max}\right)\right)\eta_k. \tag{20}$$

Substituting this results to (9) we get

$$\sum_{\mathcal{C}\in\mathcal{C}}\sum_{i\in\mathcal{C}}\sum_{t=1}^{T} \mathsf{P}\left(A_i(t) = k\right) \le \sum_{\mathcal{C}\in\mathcal{C}}(\eta_k + 2|\mathcal{C}|) + \sum_{\mathcal{C}\in\mathcal{C}}\left(|\mathcal{C}| - 1 - p\left(\sum_{j\in\mathcal{C}} p_j - p_{\max}\right)\right)\eta_k \tag{21}$$

$$+ \sum_{\mathcal{C}\in\mathcal{C}}\sum_{i\in\mathcal{C}}\sum_{t>\bar{\tau}_{k,\mathcal{C}}^i}^{T-1} \mathsf{P}\left(A_i(t+1) = k, N_k^i(t) > \eta_k\right). \tag{22}$$

Thus from Lemma 1 and (22) we have

$$\sum_{\mathcal{C}\in\mathcal{C}}\sum_{i\in\mathcal{C}}\sum_{t=1}^{T} \mathsf{P}\left(A_i(t) = k\right) \tag{23}$$

$$\le \sum_{\mathcal{C}\in\mathcal{C}} \eta_k + 2N + \sum_{\mathcal{C}\in\mathcal{C}}\left(|\mathcal{C}| - 1 - p\left(\sum_{j\in\mathcal{C}} p_j - p_{\max}\right)\right)\eta_k \tag{24}$$

$$+ \sum_{\mathcal{C}\in\mathcal{C}}\sum_{i\in\mathcal{C}}\sum_{t>\tau_{k,\mathcal{C}}}^{T-1} \left[\mathsf{P}\left(\widehat{\mu}_1^i(t) \le \mu_1 - C_1^i(t)\right) + \mathsf{P}\left(\widehat{\mu}_k^i(t) \ge \mu_k + C_k^i(t)\right)\right] \tag{25}$$

$$\stackrel{(a)}{=} \bar{\chi}(G)\eta_k + \left(N - \sum_{i=1}^{N} p_i p - \mathcal{X}(G)(1 - p_{\max}p)\right)\eta_k + 2N \tag{26}$$

$$+ \sum_{i=1}^{N}\sum_{t>\tau_{k,\mathcal{C}}}^{T-1} \left[\mathsf{P}\left(\widehat{\mu}_1^i(t) \le \mu_1 - C_1^i(t)\right) + \mathsf{P}\left(\widehat{\mu}_k^i(t) \ge \mu_k + C_k^i(t)\right)\right] \tag{27}$$

$$\le \left(\sum_{i=1}^{N}(1 - p_i p) + \bar{\chi}(G)p_{\max}p\right)\eta_k + 2N \tag{28}$$

$$+ \sum_{i=1}^{N}\sum_{t=1}^{T-1} \left[\mathsf{P}\left(\widehat{\mu}_1^i(t) \le \mu_1 - C_1^i(t)\right) + \mathsf{P}\left(\widehat{\mu}_k^i(t) \ge \mu_k + C_k^i(t)\right)\right], \tag{29}$$

where $(a)$ follows from the fact that clique covering is non overlapping. This concludes the proof of Lemma 2. □

**Lemma 3.** *Let $d_i(G)$ be the degree of agent $i$ in graph $G$. For any $\sigma_k > 0$ some constant $\zeta > 1$*

$$\mathsf{P}\left(\left|\widehat{\mu}_k^i(t) - \mu_k\right| > \sigma_k\sqrt{\frac{2(\xi+1)\log t}{N_k^i(t)}}\right) \le \frac{\log((d_i(G)+1)t)}{\log\zeta}\frac{1}{t^{(\xi+1)\left(1 - \frac{(\zeta-1)^2}{16}\right)}}. \tag{30}$$

*Proof.* For all $k$ let $X_k^i(t)$ for all $i, t$ be iid copies of $X_k$. Then we have $X_t^i\mathbf{1}\{A_i(t) = k\} = X_k^i(t)\mathbf{1}\{A_i(t) = k\}$. Recall that reward distribution of arm $k$ has mean $\mu_k$ and variance proxy $\sigma_k$. Thus $\forall i, t$ we have

$$\mathbb{E}\left(\exp\left(\lambda\left(X_k^i(t) - \mu_k\right)\right)\right) \le \exp\left(\frac{\lambda^2\sigma_k^2}{2}\right). \tag{31}$$

Define local history at every agent $i$ as follows

$$\mathcal{H}_t^i := \sigma\left(X_\tau^i, A_i(\tau), X_\tau^j\mathbf{1}\{(i,j) \in E_\tau\}, A_j(\tau)\mathbf{1}\{(i,j) \in E_\tau\}, \forall \tau \in [t], j \in \mathcal{N}_i(G)\right). \tag{32}$$

Since $\mathbf{1}\{A_j(\tau) = k\}\mathbf{1}\{(i,j) \in E_\tau\}$ for $j \in \mathcal{N}_i(G)$ is a $\mathcal{H}^i_{\tau-1}$ measurable random variable, we have

$$\mathbb{E}\left(\exp\left(\lambda\left(X^j_\tau - \mu_k\right)\mathbf{1}\{A_j(\tau) = k\}\mathbf{1}\{(i,j) \in E_\tau\}\right)\Big|\mathcal{H}^i_{\tau-1}\right) \tag{33}$$

$$= \mathbb{E}\left(\exp\left(\lambda\left(X^j_k(\tau) - \mu_k\right)\mathbf{1}\{A_j(\tau) = k\}\mathbf{1}\{(i,j) \in E_\tau\}\right)\Big|\mathcal{H}^i_{\tau-1}\right) \tag{34}$$

$$\leq \exp\left(\frac{\lambda^2\sigma^2_k}{2}\mathbf{1}\{A_j(\tau) = k\}\mathbf{1}\{(i,j) \in E_\tau\}\right). \tag{35}$$

Define a new random variable such that $\forall \tau > 0$.

$$Y^i_k(\tau) = \sum_{j=1}^N \left(X^j_k(\tau)\mathbf{1}\{A_j(\tau) = k\}\mathbf{1}\{(i,j) \in E_\tau\} - \mathbb{E}\left[X^j_k(\tau)\mathbf{1}\{A_j(\tau) = k\}\mathbf{1}\{(i,j) \in E_\tau\}\Big|\mathcal{H}^i_{\tau-1}\right]\right) \tag{36}$$

$$= \sum_{j=1}^N \left(X^j_k(\tau) - \mu_k\right)\mathbf{1}\{A_j(\tau) = k\}\mathbf{1}\{(i,j) \in E_\tau\}. \tag{37}$$

Note that $\mathbb{E}\left(Y^i_k(\tau)\right) = \mathbb{E}\left(Y^i_k(\tau)|\mathcal{H}^i_{\tau-1}\right) = 0$. Let $Z^i_k(t) = \sum_{\tau=1}^t Y^i_k(\tau)$. For any $\lambda > 0$

$$\mathbb{E}\left(\exp(\lambda Y^i_k(\tau))|\mathcal{H}^i_{\tau-1}\right) \tag{38}$$

$$= \mathbb{E}\left(\exp\left(\lambda\sum_{j=1}^N \left(X^j_k(\tau) - \mu_k\right)\mathbf{1}\{A_j(\tau) = k\}\mathbf{1}\{(i,j) \in E_\tau\}\right)\Bigg|\mathcal{H}^i_{\tau-1}\right) \tag{39}$$

$$= \mathbb{E}\left(\prod_{j=1}^N \exp\left(\lambda\left(X^j_k(\tau) - \mu_k\right)\mathbf{1}\{A_j(\tau) = k\}\mathbf{1}\{(i,j) \in E_\tau\}\right)\Bigg|\mathcal{H}^i_{\tau-1}\right) \tag{40}$$

$$\overset{(a)}{=} \prod_{j=1}^N \mathbb{E}\left(\exp\left(\lambda\left(X^j_k(\tau) - \mu_k\right)\mathbf{1}\{A_j(\tau) = k\}\mathbf{1}\{(i,j) \in E_\tau\}\right)\Big|\mathcal{H}^i_{\tau-1}\right) \tag{41}$$

$$\leq \prod_{j=1}^N \exp\left(\frac{\lambda^2\sigma^2_k}{2}\mathbf{1}\{A_j(\tau) = k\}\mathbf{1}\{(i,j) \in E_\tau\}\right) \tag{42}$$

$$= \exp\left(\frac{\lambda^2\sigma^2_k}{2}\sum_{j=1}^N \mathbf{1}\{A_j(\tau) = k\}\mathbf{1}\{(i,j) \in E_\tau\}\right). \tag{43}$$

Equality $(a)$ follows from the fact that random variables $\left\{\exp\left(\lambda\left(X^j_k(\tau) - \mu_k\right)\mathbf{1}\{A_j(\tau) = k\}\mathbf{1}\{(i,j) \in E_\tau\}\right)\right\}_{j=1}^N$ are conditionally independent with respect to $\mathcal{H}^i_{\tau-1}$. Since $\mathbf{1}\{A_j(\tau) = k\}, \mathbf{1}\{(i,j) \in E_\tau\}$ are $\mathcal{H}^i_{\tau-1}$ measurable, and so

$$\mathbb{E}\left(\exp\left(\lambda Y^i_k(\tau) - \frac{\lambda^2\sigma^2_k}{2}\sum_{j=1}^N \mathbf{1}\{A_j(\tau) = k\}\mathbf{1}\{(i,j) \in E_\tau\}\right)\Bigg||\mathcal{H}^i_{\tau-1}\right) \leq 1. \tag{44}$$

Let $N^i_k(t) = \sum_{\tau=1}^t \sum_{j=1}^N \mathbf{1}\{A_i(\tau) = k\}\mathbf{1}\{(i,j) \in E_\tau\}$. Further, using the tower property of conditional expectation we have

$$\mathbb{E}\left(\exp\left(\lambda Z^i_k(t) - \frac{\lambda^2\sigma^2_k}{2}N^i_k(t)\right)\Big|\mathcal{H}^i_{t-1}\right) \leq \exp\left(\lambda Z^i_k(t-1) - \frac{\lambda^2\sigma^2_k}{2}N^i_k(t-1)\right). \tag{45}$$

Repeating the above step $t$ times we have

$$\mathbb{E}\left(\exp\left(\lambda Z^i_k(t) - \frac{\lambda^2\sigma^2_k}{2}N^i_k(t)\right)\right) \leq 1. \tag{46}$$

Note that we have

$$\mathsf{P}\left(\exp\left(\lambda Z^i_k(t) - \frac{\lambda^2\sigma^2_i}{2}N^i_k(t)\right) \geq \exp\left(2\kappa\vartheta\right)\right) \tag{47}$$

$$= \mathsf{P}\left(\lambda Z_k^i(t) - \frac{\lambda^2 \sigma_k^2}{2} N_k^i(t) \geq 2\kappa\vartheta\right) \tag{48}$$

$$= \mathsf{P}\left(\frac{Z_k^i(t)}{\sqrt{N_k^i(t)}} \geq \frac{2\kappa\vartheta}{\lambda\sqrt{N_k^i(t)}} + \frac{\sigma_k^2}{2}\lambda\sqrt{N_k^i(t)}\right). \tag{49}$$

Fix a constant $\zeta > 1$. Then $1 \leq N_k^i(t) \leq \zeta^{D_t}$ where $D_t = \frac{\log((d_i(G)+1)t)}{\log\zeta}$. For $\lambda_l = \frac{2}{\sigma_k}\sqrt{\frac{\kappa\vartheta}{\zeta^{l-1/2}}}$ and $\zeta^{l-1} \leq N_k^i(t) \leq \zeta^l$ we have

$$\frac{2\kappa\vartheta}{\lambda_l}\sqrt{\frac{1}{N_k^i(t)}} + \frac{\sigma_k^2}{2}\lambda_l\sqrt{N_k^i(t)} = \sigma_k\sqrt{\kappa\vartheta}\left(\sqrt{\frac{\zeta^{l-1/2}}{N_k^i(t)}} + \sqrt{\frac{N_k^i(t)}{\zeta^{l-1/2}}}\right) \leq \sqrt{\vartheta}, \tag{50}$$

where $\kappa = \frac{1}{\sigma_k^2\left(\zeta^{\frac{1}{4}} + \zeta^{-\frac{1}{4}}\right)^2}$.

Then we have

$$\left\{\frac{Z_k^i(t)}{\sqrt{N_k^i(t)}} \geq \sqrt{\vartheta}\right\} \subset \cup_{l=1}^{D_t}\left\{\frac{Z_k^i(t)}{\sqrt{N_k^i(t)}} \geq \frac{2\kappa\vartheta}{\lambda_l\sqrt{N_k^i(t)}} + \frac{\sigma_k^2}{2}\lambda_l\sqrt{N_k^i(t)}\right\} \tag{51}$$

$$= \cup_{l=1}^{D_t}\left\{\lambda_l Z_k^i(t) - \frac{\lambda_l^2\sigma_k^2}{2}N_k^i(t) \geq 2\kappa\vartheta\right\}. \tag{52}$$

Recall from the Markov inequality that $\mathsf{P}(Y \geq a) \leq \frac{\mathbb{E}(Y)}{a}$ for any positive random variable $Y$. Thus from (52) and Markov inequality we get,

$$\mathsf{P}\left(\frac{Z_k^i(t)}{\sqrt{N_k^i(t)}} \geq \sqrt{\vartheta}\right) \leq \sum_{l=1}^{D_t}\exp(-2\kappa\vartheta). \tag{53}$$

Then we have,

$$\mathsf{P}\left(\frac{Z_k^i(t)}{N_k^i(t)} \geq \sqrt{\frac{\vartheta}{N_k^i(t)}}\right) \leq \sum_{l=1}^{D_t}\exp(-2\kappa\vartheta) \tag{54}$$

Substituting $\vartheta = 2\sigma_k^2(\xi+1)\log t$ we get

$$\mathsf{P}\left(\left|\widehat{\mu}_k^i(t) - \mu_k\right| > \sigma_k\sqrt{\frac{2(\xi+1)\log t}{N_k^i(t)}}\right) \leq \frac{\log((d_i(G)+1)t)}{\log\zeta}\exp\left(-\frac{4(\xi+1)\log t}{\left(\zeta^{\frac{1}{4}} + \zeta^{-\frac{1}{4}}\right)^2}\right). \tag{55}$$

Note that $\forall \zeta > 1$ we have

$$\frac{4}{\left(\zeta^{\frac{1}{4}} + \zeta^{-\frac{1}{4}}\right)^2} \geq 1 - \frac{(\zeta-1)^2}{16} \tag{56}$$

Then we have

$$\mathsf{P}\left(\left|\widehat{\mu}_k^i(t) - \mu_k\right| > \sigma_k\sqrt{\frac{2(\xi+1)\log t}{N_k^i(t)}}\right) \leq \frac{\log((d_i(G)+1)t)}{\log\zeta}\frac{1}{t^{(\xi+1)\left(1-\frac{(\zeta-1)^2}{16}\right)}}. \tag{57}$$

This concludes the proof of Lemma 3. $\qquad\square$

**Lemma 4.** *Let $\zeta = 1.3, \xi \geq 1.1, d_i \geq 0$ and $t \in [T]$. Then we have*

$$\sum_{t=1}^{T-1}\frac{1}{\log\zeta}\frac{\log((d_i+1)t)}{t^{(\xi+1)\left(1-\frac{(\zeta-1)^2}{16}\right)}} \leq 12\log(3(d_i+1)) + 3(\log(d_i+1)+1) \tag{58}$$

*Proof.* For $\zeta = 1.3$ we have $\frac{1}{\log \zeta} < 8.78$. Further $(\xi + 1)\left(1 - \frac{(\zeta-1)^2}{16}\right) > 2$ and $\forall t \geq 3$ we see that $\frac{\log((d_i+1)t)}{t^{(\xi+1)\left(1-\frac{(\zeta-1)^2}{16}\right)}}$ is monotonically decreasing. Thus we have

$$\sum_{t=1}^{T-1} \frac{\log((d_i+1)t)}{t^{(\xi+1)\left(1-\frac{(\zeta-1)^2}{16}\right)}} \leq 1.362 \log(3(d_i+1)) + \int_3^{T-1} \frac{\log((d_i+1)t)}{t^2} dt \tag{59}$$

Let $z = (d_i+1)t$. Then we have

$$\int_3^{T-1} \frac{\log((d_i+1)t)}{t^2} dt = (d_i+1) \int_{3(d_i+1)}^{(d_i+1)(T-1)} \frac{\log z}{z^2} dz \tag{60}$$

$$= (d_i+1) \left[ -\frac{\log z}{z} - \frac{1}{z} \right]_{3((d_i+1)}^{(d_i+1)(T-1)} \tag{61}$$

Thus we have

$$\int_3^{T-1} \frac{\log((d_i+1)t)}{t^2} dt \leq (d_i+1) \left[ \frac{\log(d_i+1)}{3(d_i+1)} + \frac{1}{3(d_i+1)} \right] \tag{62}$$

$$= \frac{1}{3} \log(d_i+1) + \frac{1}{3} \tag{63}$$

Recall that For $\zeta = 1.3$ we have $\frac{1}{\log \zeta} < 8.78$. Thus the proof of Lemma 4 follows from (59) and (63). $\qquad \square$

Now we prove Theorem 1 as follows. Recall that group regret can be given as $\text{Reg}_G(T) = \sum_{i=1}^N \sum_{k>1} \Delta_k \cdot \mathbb{E}\left[n_k^i(t)\right]$. Thus using Lemmas 2, 3 and 4 we obtain

$$\text{Reg}_G(T) \leq 8(\xi+1)\sigma_k^2 \left( \sum_{i=1}^N (1 - p_i p) + \bar{\chi}(G) p_{\max} p \right) \left( \sum_{k>1} \frac{\log T}{\Delta_k} \right) \tag{64}$$

$$+ 5N \sum_{k>1} \Delta_k + 4 \sum_{i=1}^N \left( 3\log(3(d_i(G)+1)) + (\log(d_i(G)+1)) \right) \sum_{k>1} \Delta_k \tag{65}$$

## B  Proof of Theorem 2

In this section we consider the case where agents pass messages up to $\gamma$ hop neighbors with each hop adding a delay of 1 time step. Let $\mathcal{C}_\gamma$ be a non overlapping clique covering of $G_\gamma$. For any $\mathcal{C} \in \mathcal{C}_\gamma$ and $i,j \in \mathcal{C}$ let $\gamma_i = \max_{j \in \mathcal{C}} d(i,j)$ be the maximum distance (in graph $G$) between agent $i$ and any other agent $j$ in the same clique in graph $G_\gamma$. Let $\mathbf{1}\{(i,j) \in E_{\tau',\tau}\}$ is a random variable that takes value 1 if at time $\tau$ agent $i$ receives the message initiated by agent $j$ at time $\tau'$. Recall that each communicated message fails with probability $1 - p$ and each agent $i$ incorporates the messages it receives from its neighbors with probability $p_i$.

We follow an approach similar to proof of Theorem 1. We star by providing a tail probability bound similar to Lemma 3.

**Lemma 5.** *Let $d_i(G_\gamma)$ be the degree of agent $i$ in graph $G_\gamma$. For any $\sigma_k > 0$ some constant $\zeta > 1$*

$$P\left( \left|\widehat{\mu}_k^i(t) - \mu_k\right| > \sigma_k \sqrt{\frac{2(\xi+1)\log t}{N_k^i(t)}} \right) \leq \frac{\log((d_i(G_\gamma)+1)t)}{\log \zeta} \frac{1}{t^{(\xi+1)\left(1-\frac{(\zeta-1)^2}{16}\right)}}. \tag{66}$$

*Proof.* For all $k$ let $X_k^i(t)$ for all $i,t$ be iid copies of $X_k$. Then we have $X_k^i \mathbf{1}\{A_i(t) = k\} = X_k^i(t)\mathbf{1}\{A_i(t) = k\}$. Recall that reward distribution of arm $k$ has mean $\mu_k$ and variance proxy $\sigma_k$. Thus $\forall i, t$ we have

$$\mathbb{E}\left(\exp\left(\lambda\left(X_k^i(t) - \mu_k\right)\right)\right) \leq \exp\left(\frac{\lambda^2 \sigma_k^2}{2}\right). \tag{67}$$

Define local history at every agent $i$ as follows

$$\mathcal{H}_t^i := \sigma\left(X_{\tau'}^i, A_i(\tau'), X_{\tau'}^j \mathbf{1}\{(i,j) \in E_{\tau',\tau}\}, A_j(\tau')\mathbf{1}\{(i,j) \in E_{\tau',\tau}\}, \forall \tau', \tau \in [t], j \in \mathcal{N}_i(G_\gamma)\right). \tag{68}$$

Since $\mathbf{1}\{A_j(\tau') = k\}\mathbf{1}\{(i,j) \in E_{\tau',\tau}\}$ for $j \in \mathcal{N}_i(G_\gamma)$ is a $\mathcal{H}_{\tau-1}^i$ measurable random variable, we have $\forall \tau' \leq \tau$

$$\mathbb{E}\left(\exp\left(\lambda\left(X_{\tau'}^j - \mu_k\right)\mathbf{1}\{A_j(\tau') = k\}\mathbf{1}\{(i,j) \in E_{\tau',\tau}\}\right)\Big| \mathcal{H}_{\tau-1}^i\right) \tag{69}$$

$$= \mathbb{E}\left(\exp\left(\lambda\left(X_k^j(\tau') - \mu_k\right)\mathbf{1}\{A_j(\tau') = k\}\mathbf{1}\{(i,j) \in E_{\tau',\tau}\}\right)\Big| \mathcal{H}_{\tau-1}^i\right) \tag{70}$$

$$\leq \exp\left(\frac{\lambda^2\sigma_k^2}{2}\mathbf{1}\{A_j(\tau') = k\}\mathbf{1}\{(i,j) \in E_{\tau',\tau}\}\right). \tag{71}$$

Define a new random variable such that $\forall \tau > 0$ and $\tau' \leq \tau$

$$Y_k^i(\tau) = \sum_{j=1}^N \sum_{\tau'=1}^\tau \left(X_k^j(\tau')\mathbf{1}\{A_j(\tau') = k\}\mathbf{1}\{(i,j) \in E_{\tau',\tau}\}\right. \tag{72}$$

$$\left. - \mathbb{E}\left[X_k^j(\tau')\mathbf{1}\{A_j(\tau') = k\}\mathbf{1}\{(i,j) \in E_{\tau',\tau}\}\Big| \mathcal{H}_{\tau-1}^i\right]\right) \tag{73}$$

$$= \sum_{j=1}^N \sum_{\tau'=1}^\tau \left(X_k^j(\tau') - \mu_k\right)\mathbf{1}\{A_j(\tau') = k\}\mathbf{1}\{(i,j) \in E_{\tau',\tau}\}. \tag{74}$$

Note that $\mathbb{E}\left(Y_k^i(\tau)\right) = \mathbb{E}\left(Y_k^i(\tau)|\mathcal{H}_{\tau-1}^i\right) = 0$. Let $Z_k^i(t) = \sum_{\tau=1}^t Y_k^i(\tau)$. For any $\lambda > 0$

$$\mathbb{E}\left(\exp(\lambda Y_k^i(\tau))|\mathcal{H}_{\tau-1}^i\right) \tag{75}$$

$$= \mathbb{E}\left(\exp\left(\lambda \sum_{j=1}^N \sum_{\tau'=1}^\tau \left(X_k^j(\tau') - \mu_k\right)\mathbf{1}\{A_j(\tau') = k\}\mathbf{1}\{(i,j) \in E_{\tau',\tau}\}\right)\Big| \mathcal{H}_{\tau-1}^i\right) \tag{76}$$

$$= \mathbb{E}\left(\prod_{j=1}^N \prod_{\tau'=1}^\tau \exp\left(\lambda\left(X_k^j(\tau') - \mu_k\right)\mathbf{1}\{A_j(\tau') = k\}\mathbf{1}\{(i,j) \in E_{\tau',\tau}\}\right)\Big| \mathcal{H}_{\tau-1}^i\right) \tag{77}$$

$$\overset{(a)}{=} \prod_{j=1}^N \prod_{\tau'=1}^\tau \mathbb{E}\left(\exp\left(\lambda\left(X_k^j(\tau') - \mu_k\right)\mathbf{1}\{A_j(\tau') = k\}\mathbf{1}\{(i,j) \in E_{\tau',\tau}\}\right)\Big| \mathcal{H}_{\tau-1}^i\right) \tag{78}$$

$$\leq \prod_{j=1}^N \prod_{\tau'=1}^\tau \exp\left(\frac{\lambda^2\sigma_k^2}{2}\mathbf{1}\{A_j(\tau') = k\}\mathbf{1}\{(i,j) \in E_{\tau',\tau}\}\right) \tag{79}$$

$$= \exp\left(\frac{\lambda^2\sigma_k^2}{2}\sum_{j=1}^N \sum_{\tau'=1}^\tau \mathbf{1}\{A_j(\tau') = k\}\mathbf{1}\{(i,j) \in E_{\tau',\tau}\}\right). \tag{80}$$

Equality $(a)$ follows from the fact that $\forall \tau' \leq \tau$ random variables $\left\{\exp\left(\lambda\left(X_k^j(\tau') - \mu_k\right)\mathbf{1}\{A_j(\tau') = k\}\mathbf{1}\{(i,j) \in E_{\tau'\tau}\}\right)\right\}_{j=1}^N$ are conditionally independent with respect to $\mathcal{H}_{\tau-1}^i$. Since $\mathbf{1}\{A_j(\tau') = k\}, \mathbf{1}\{(i,j) \in E_{\tau',\tau}\}$ are $\mathcal{H}_{\tau-1}^i$ measurable, and so

$$\mathbb{E}\left(\exp\left(\lambda Y_k^i(\tau) - \frac{\lambda^2\sigma_k^2}{2}\sum_{j=1}^N \sum_{\tau'=1}^\tau \mathbf{1}\{A_j(\tau') = k\}\mathbf{1}\{(i,j) \in E_{\tau',\tau}\}\right)\Big| \mathcal{H}_{\tau-1}^i\right) \leq 1. \tag{81}$$

Let $N_k^i(t) = \sum_{\tau=1}^t \sum_{\tau'=1}^\tau \sum_{j=1}^N \mathbf{1}\{A_i(\tau') = k\}\mathbf{1}\{(i,j) \in E_{\tau',\tau}\}$. Further, using the tower property of conditional expectation we have

$$\mathbb{E}\left(\exp\left(\lambda Z_k^i(t) - \frac{\lambda^2\sigma_k^2}{2}N_k^i(t)\right)\Big| \mathcal{H}_{t-1}^i\right) \leq \exp\left(\lambda Z_k^i(t-1) - \frac{\lambda^2\sigma_k^2}{2}N_k^i(t-1)\right). \tag{82}$$

Repeating the above step $t$ times we have

$$\mathbb{E}\left(\exp\left(\lambda Z_k^i(t) - \frac{\lambda^2\sigma_k^2}{2}N_k^i(t)\right)\right) \leq 1. \tag{83}$$

Note that we have

$$\mathsf{P}\left(\exp\left(\lambda Z_k^i(t) - \frac{\lambda^2\sigma_i^2}{2}N_k^i(t)\right) \geq \exp\left(2\kappa\vartheta\right)\right) \tag{84}$$

$$= \mathsf{P}\left(\lambda Z_k^i(t) - \frac{\lambda^2\sigma_k^2}{2}N_k^i(t) \geq 2\kappa\vartheta\right) \tag{85}$$

$$= \mathsf{P}\left(\frac{Z_k^i(t)}{\sqrt{N_k^i(t)}} \geq \frac{2\kappa\vartheta}{\lambda\sqrt{N_k^i(t)}} + \frac{\sigma_k^2}{2}\lambda\sqrt{N_k^i(t)}\right). \tag{86}$$

Fix a constant $\zeta > 1$. Then $1 \leq N_k^i(t) \leq \zeta^{D_t}$ where $D_t = \frac{\log((d_i(G_\gamma)+1)t)}{\log\zeta}$. For $\lambda_l = \frac{2}{\sigma_k}\sqrt{\frac{\kappa\vartheta}{\zeta^{l-1/2}}}$ and $\zeta^{l-1} \leq N_k^i(t) \leq \zeta^l$ we have

$$\frac{2\kappa\vartheta}{\lambda_l}\sqrt{\frac{1}{N_k^i(t)}} + \frac{\sigma_k^2}{2}\lambda_l\sqrt{N_k^i(t)} = \sigma_k\sqrt{\kappa\vartheta}\left(\sqrt{\frac{\zeta^{l-1/2}}{N_k^i(t)}} + \sqrt{\frac{N_k^i(t)}{\zeta^{l-1/2}}}\right) \leq \sqrt{\vartheta}, \tag{87}$$

where $\kappa = \frac{1}{\sigma_k^2\left(\zeta^{\frac{1}{4}}+\zeta^{-\frac{1}{4}}\right)^2}$.

Then we have

$$\left\{\frac{Z_k^i(t)}{\sqrt{N_k^i(t)}} \geq \sqrt{\vartheta}\right\} \subset \cup_{l=1}^{D_t}\left\{\frac{Z_k^i(t)}{\sqrt{N_k^i(t)}} \geq \frac{2\kappa\vartheta}{\lambda_l\sqrt{N_k^i(t)}} + \frac{\sigma_k^2}{2}\lambda_l\sqrt{N_k^i(t)}\right\} \tag{88}$$

$$= \cup_{l=1}^{D_t}\left\{\lambda_l Z_k^i(t) - \frac{\lambda_l^2\sigma_k^2}{2}N_k^i(t) \geq 2\kappa\vartheta\right\}. \tag{89}$$

Recall from the Markov inequality that $\mathsf{P}(Y \geq a) \leq \frac{\mathbb{E}(Y)}{a}$ for any positive random variable $Y$. Thus from (89) and Markov inequality we get,

$$\mathsf{P}\left(\frac{Z_k^i(t)}{\sqrt{N_k^i(t)}} \geq \sqrt{\vartheta}\right) \leq \sum_{l=1}^{D_t}\exp(-2\kappa\vartheta). \tag{90}$$

Then we have,

$$\mathsf{P}\left(\frac{Z_k^i(t)}{N_k^i(t)} \geq \sqrt{\frac{\vartheta}{N_k^i(t)}}\right) \leq \sum_{l=1}^{D_t}\exp(-2\kappa\vartheta) \tag{91}$$

Recall that $\forall\zeta > 1$ we have

$$\frac{4}{\left(\zeta^{\frac{1}{4}}+\zeta^{-\frac{1}{4}}\right)^2} \geq 1 - \frac{(\zeta-1)^2}{16} \tag{92}$$

Substituting $\vartheta = 2\sigma_k^2(\xi+1)\log t$ we get

$$\mathsf{P}\left(\left|\widehat{\mu}_k^i(t) - \mu_k\right| > \sigma_k\sqrt{\frac{2(\xi+1)\log t}{N_k^i(t)}}\right) \leq \frac{\log((d_i(G_\gamma)+1)t)}{\log\zeta}\frac{1}{t^{(\xi+1)\left(1-\frac{(\zeta-1)^2}{16}\right)}}. \tag{93}$$

This concludes the proof of Lemma 5. □

We prove a Lemma similar to Lemma 2 for message-passing as follows.

**Lemma 6.** *Let* $\bar{\chi}(G_\gamma)$ *is the clique number of graph* $G_\gamma$. *Let* $\eta_k = \left(\frac{8(\xi+1)\sigma_k^2}{\Delta_k^2}\right) \log T$. *Then we have*

$$\sum_{i=1}^{N} \mathbb{E}[n_k^i(T)] \leq \left(\sum_{i=1}^{N}(1 - p_i p^{\gamma_i}) + \bar{\chi}(G_\gamma) \max_{i \in [N]} p_i p^{\gamma_i}\right) \eta_k + N(\gamma+1) + \tag{94}$$

$$+ \sum_{i=1}^{N} \sum_{t=1}^{T-1} \left[\mathsf{P}\left(\widehat{\mu}_1^i(t) \leq \mu_1 - C_1^i(t)\right) + \mathsf{P}\left(\widehat{\mu}_k^i(t) \geq \mu_k + C_k^i(t)\right)\right] \tag{95}$$

*Proof.* Note that for each suboptimal arm $k > 1$ we have

$$\sum_{i=1}^{N} \mathbb{E}[n_k^i(T)] = \sum_{i=1}^{N} \sum_{t=1}^{T} \mathsf{P}\left(A_i(t) = k\right) = \sum_{\mathcal{C} \in \mathcal{C}_\gamma} \sum_{i \in \mathcal{C}} \sum_{t=1}^{T} \mathsf{P}\left(A_i(t) = k\right). \tag{96}$$

Let $\tau_{k,\mathcal{C}}$ denote the maximum time step when the total number of times arm $k$ has been played by all the agents in clique $\mathcal{C}$ is at most $\eta_k + |\mathcal{C}|$ times. This can be stated as $\tau_{k,\mathcal{C}} := \max\{t \in [T] : \sum_{i \in \mathcal{C}} n_k^i(t) \leq \eta_k + |\mathcal{C}|\}$. Then, we have that $\eta_k < \sum_{i \in \mathcal{C}} n_k^i(\tau_{k,\mathcal{C}}) \leq \eta_k + |\mathcal{C}|$.

For each agent $i \in \mathcal{C}$ let

$$\bar{N}_k^i(t) := \sum_{j \in \mathcal{C}} \sum_{\tau=1}^{t} \sum_{\tau'=1}^{\tau} \mathbf{1}\{A_j(\tau') = k\}\mathbf{1}\{(i,j) \in E_{\tau',\tau}\},$$

denote the sum of the total number of times agent $i$ pulled arm $k$ and the total number of observations it received from agents in its clique about arm $k$ until time $t$. Define $\bar{\tau}_{k,\mathcal{C}}^i := \max\{t \in [T] : \bar{N}_k^i(t) \leq \eta_k\}$. For each agent $i \in [N]$ let $\bar{\tau}_{k,\mathcal{C}}^i = \max\{\tau_{k,\mathcal{C}} + \gamma_i - 1, \bar{\tau}_{k,\mathcal{C}}^i\}$.

Note that $N_k^i(t) \geq \bar{N}_k^i(t), \forall t$, hence for all $i \in \mathcal{C}$ we have $N_k^i(t) > \eta_k, \forall t > \bar{\tau}_{k,\mathcal{C}}^i$. Here we consider that $\bar{\tau}_{k,\mathcal{C}}^i \geq \tau_{k,\mathcal{C}}, \forall i$. From regret results it follows that regret for this case is greater than the regret for the case where $\bar{\tau}_{k,\mathcal{C}}^i < \tau_{k,\mathcal{C}}$ for some (or all) $i$.

We analyse the expected number of times agents pull suboptimal arm $k$ as follows,

$$\sum_{\mathcal{C} \in \mathcal{C}_\gamma} \sum_{i \in \mathcal{C}} \sum_{t=1}^{T} \mathbf{1}\{A_i(t) = k\} \tag{97}$$

$$= \sum_{\mathcal{C} \in \mathcal{C}_\gamma} \sum_{i \in \mathcal{C}} \sum_{t=1}^{\tau_{k,\mathcal{C}}} \mathbf{1}\{A_i(t) = k\} + \sum_{\mathcal{C} \in \mathcal{C}_\gamma} \sum_{i \in \mathcal{C}} \sum_{t > \tau_{k,\mathcal{C}}}^{\bar{\tau}_{k,\mathcal{C}}^i} \mathbf{1}\{A_i(t) = k\} + \sum_{\mathcal{C} \in \mathcal{C}_\gamma} \sum_{i \in \mathcal{C}} \sum_{t > \bar{\tau}_{k,\mathcal{C}}^i}^{T} \mathbf{1}\{A_i(t) = k\} \tag{98}$$

$$\leq \sum_{\mathcal{C} \in \mathcal{C}_\gamma}(\eta_k + |\mathcal{C}|) + \sum_{\mathcal{C} \in \mathcal{C}_\gamma} \sum_{i \in \mathcal{C}} \sum_{t > \tau_{k,\mathcal{C}}}^{\bar{\tau}_{k,\mathcal{C}}^i} \mathbf{1}\{A_i(t) = k\} \tag{99}$$

$$+ \sum_{\mathcal{C} \in \mathcal{C}_\gamma} \sum_{i \in \mathcal{C}} \sum_{t > \bar{\tau}_{k,\mathcal{C}}^i}^{T} \mathbf{1}\{A_i(t) = k\}\mathbf{1}\{N_k^i(t-1) > \eta_k\}. \tag{100}$$

Taking expectation we have

$$\sum_{\mathcal{C} \in \mathcal{C}_\gamma} \sum_{i \in \mathcal{C}} \sum_{t=1}^{T} \mathsf{P}\left(A_i(t) = k\right) \tag{101}$$

$$\leq \sum_{\mathcal{C} \in \mathcal{C}_\gamma}(\eta_k + 2|\mathcal{C}|) + \sum_{\mathcal{C} \in \mathcal{C}_\gamma} \sum_{i \in \mathcal{C}} \sum_{t > \tau_{k,\mathcal{C}}}^{\bar{\tau}_{k,\mathcal{C}}^i} \mathsf{P}\left(A_i(t) = k\right) \tag{102}$$

$$+ \sum_{\mathcal{C} \in \mathcal{C}_\gamma} \sum_{i \in \mathcal{C}} \sum_{t > \bar{\tau}_{k,\mathcal{C}}^i}^{T-1} \mathsf{P}\left(A_i(t+1) = k, N_k^i(t) > \eta_k\right). \tag{103}$$

**Case 1**. For agent $i$ we have that $\tau_{k,\mathcal{C}} + \gamma_i - 1 \geq \bar{\tau}^i_{k,\mathcal{C}}$ then we have $\bar{\bar{\tau}}^i_{k,\mathcal{C}} = \tau_{k,\mathcal{C}} + \gamma_i - 1$. Then we have $\sum_{t > \tau_{k,\mathcal{C}}}^{\bar{\tau}^i_{k,\mathcal{C}}} \mathbf{1}\{A_i(t) = k\} \leq \gamma_i - 1$

**Case 2**. For agent $i$ we have that $\tau_{k,\mathcal{C}} + \gamma_i - 1 < \bar{\tau}^i_{k,\mathcal{C}}$ then we have $\bar{\bar{\tau}}^i_{k,\mathcal{C}} = \bar{\tau}^i_{k,\mathcal{C}}$.

$$\sum_{t > \tau_{k,\mathcal{C}}}^{\bar{\bar{\tau}}^i_{k,\mathcal{C}}} \mathbf{1}\{A_i(t) = k\} \tag{104}$$

$$= \tilde{N}^i_k(\bar{\bar{\tau}}^i_{k,\mathcal{C}}) - \sum_{t=1}^{\tau_{k,\mathcal{C}}} \mathbf{1}\{A_i(t) = k\} - \sum_{j \neq i, j \in \mathcal{C}} \sum_{t=1}^{\bar{\bar{\tau}}^i_{k,\mathcal{C}}} \sum_{\tau=1}^{t} \mathbf{1}\{A_j(\tau) = k\} \mathbf{1}\{(i,j) \in E_{\tau,t}\} \tag{105}$$

$$\leq \tilde{N}^i_k(\bar{\bar{\tau}}^i_{k,\mathcal{C}}) - \sum_{t=1}^{\tau_{k,\mathcal{C}}} \mathbf{1}\{A_i(t) = k\} - \sum_{j \neq i, j \in \mathcal{C}} \sum_{t=1}^{\tau_{k,\mathcal{C}}+\gamma_i-1} \sum_{\tau=1}^{t} \mathbf{1}\{A_j(\tau) = k\} \mathbf{1}\{(i,j) \in E_{\tau,t}\}. \tag{106}$$

Taking the expectation we have

$$\sum_{i \in \mathcal{C}} \sum_{t > \tau_{k,\mathcal{C}}}^{\bar{\bar{\tau}}^i_{k,\mathcal{C}}} \mathsf{P}\left(A_i(t) = k\right) \leq |\mathcal{C}|\eta_k - \eta_k + \sum_{i \in \mathcal{C}}(\gamma_i - 1) - \sum_{i \in \mathcal{C}} p_i p^{\gamma_i} \sum_{j \neq i, j \in \mathcal{C}} \sum_{t=1}^{\tau_{k,\mathcal{C}}} \mathsf{P}\left(A_j(t) = k\right) \tag{107}$$

$$= |\mathcal{C}|\eta_k - \eta_k + \sum_{i \in \mathcal{C}}(\gamma_i - 1) - \sum_{i \in \mathcal{C}} p_i p^{\gamma_i} \sum_{j \neq i, j \in \mathcal{C}} \sum_{t=1}^{\tau_{k,\mathcal{C}}} \mathbb{E}\left(n^j_k(\tau_{k,\mathcal{C}})\right) \tag{108}$$

$$\leq \left(|\mathcal{C}| - 1 - \left(\sum_{j \in \mathcal{C}} p_j p^{\gamma_j} - \max_{i \in [N]} p_i p^{\gamma_i}\right)\right) \eta_k + \sum_{i \in \mathcal{C}}(\gamma_i - 1). \tag{109}$$

Substituting these results to (103) we get

$$\sum_{\mathcal{C} \in \mathcal{C}_\gamma} \sum_{i \in \mathcal{C}} \sum_{t=1}^{T} \mathsf{P}\left(A_i(t) = k\right) \leq \sum_{\mathcal{C} \in \mathcal{C}_\gamma} \left(|\mathcal{C}| - 1 - \left(\sum_{j \in \mathcal{C}} p_j p^{\gamma_j} - \max_{i \in [N]} p_i p^{\gamma_i}\right)\right) \eta_k + \sum_{i \in [N]}(\gamma_i - 1) \tag{110}$$

$$+ \sum_{\mathcal{C} \in \mathcal{C}_\gamma} (\eta_k + 2|\mathcal{C}|) + \sum_{\mathcal{C} \in \mathcal{C}_\gamma} \sum_{i \in \mathcal{C}} \sum_{t > \bar{\bar{\tau}}^i_{k,\mathcal{C}}}^{T-1} \mathsf{P}\left(A_i(t+1) = k, N^i_k(t) > \eta_k\right) \tag{111}$$

$$\leq \left(\sum_{i=1}^{N}(1 - p_i p^{\gamma_i}) + \bar{\chi}(G_\gamma) \max_{i \in [N]} p_i p^{\gamma_i}\right) \eta_k + \sum_{i \in [N]} \gamma_i + N \tag{112}$$

$$+ \sum_{\mathcal{C} \in \mathcal{C}_\gamma} \sum_{i \in \mathcal{C}} \sum_{t > \bar{\bar{\tau}}^i_{k,\mathcal{C}}}^{T-1} \mathsf{P}\left(A_i(t+1) = k, N^i_k(t) > \eta_k\right) \tag{113}$$

This concludes the proof of Lemma 6. □

Now we prove Theorem 2 as follows. Thus using Lemmas 4, 5 and 6 we obtain

$$\mathsf{Reg}_G(T) \leq 8(\xi + 1)\sigma^2_k \left(\sum_{i=1}^{N}(1 - p_i p^{\gamma_i}) + \bar{\chi}(G_\gamma) \max_{i \in [N]} p_i p^{\gamma_i}\right) \left(\sum_{k>1} \frac{\log T}{\Delta_k}\right) \tag{114}$$

$$+ \left(\sum_{i=1}^{N} \gamma_i + 4N\right) \sum_{k>1} \Delta_k + 4 \sum_{i=1}^{N} \left(3\log(3(d_i(G_\gamma) + 1)) + (\log(d_i(G_\gamma) + 1))\right) \sum_{k>1} \Delta_k \tag{115}$$

## C Proof of Theorem 3

Agents receive information from their neighbors with a stochastic time delay. Let $\mathcal{N}_D$ be the maximum number of outstanding arm pulls by all the agent. We start by proving a result similar to Lemma 2.

**Lemma 7.** *Let $\bar{\chi}(G)$ is the clique number of graph $G$. Let $\eta_k = \left( \frac{8(\xi+1)\sigma_k^2}{\Delta_k^2} \right) \log T$. Then we have*

$$\sum_{i=1}^{N} \mathbb{E}[n_k^i(T)] \leq \bar{\chi}(G)\eta_k + \mathbb{E}[\mathcal{N}_D] + 2N + \tag{116}$$

$$+ \sum_{i=1}^{N} \sum_{t=1}^{T-1} \left[ \mathsf{P}\left( \widehat{\mu}_1^i(t) \leq \mu_1 - C_1^i(t) \right) + \mathsf{P}\left( \widehat{\mu}_k^i(t) \geq \mu_k + C_k^i(t) \right) \right] \tag{117}$$

*Proof.* Let $\mathcal{C}$ be a non overlapping clique covering of $G$. Note that for each suboptimal arm $k > 1$ we have

$$\sum_{i=1}^{N} \mathbb{E}[n_k^i(T)] = \sum_{i=1}^{N} \sum_{t=1}^{T} \mathsf{P}\left( A_i(t) = k \right) = \sum_{\mathcal{C} \in \mathcal{C}} \sum_{i \in \mathcal{C}} \sum_{t=1}^{T} \mathsf{P}\left( A_i(t) = k \right). \tag{118}$$

Let $\tau_{k,\mathcal{C}}$ denote the maximum time step such that the total number of arm pulls shared by agents in clique $\mathcal{C}$ from arm $k$ is at most $\eta_k + |\mathcal{C}|$. For each agent $i \in \mathcal{C}$ let $D_i(\tau_{k,\mathcal{C}})$ be the number of outstanding messages by agent $i$ from arm $k$ at time $\tau_{k,\mathcal{C}}$. This can be stated as $\tau_{k,\mathcal{C}} := \max\{t \in [T] : \sum_{i \in \mathcal{C}} n_k^i(t) \leq \eta_k + \sum_{i \in \mathcal{C}} D_i(\tau_{k,\mathcal{C}}) + |\mathcal{C}|\}$. Then, we have that $\eta_k + \sum_{i \in \mathcal{C}} D_i(\tau_{k,\mathcal{C}}) < \sum_{i \in \mathcal{C}} n_k^i(\tau_{k,\mathcal{C}}) \leq \eta_k + \sum_{i \in \mathcal{C}} D_i(\tau_{k,\mathcal{C}}) + |\mathcal{C}|$.

Note that for all $i \in \mathcal{C}$ we have $N_k^i(t) > \eta_k, t > \tau_{k,\mathcal{C}}$.

We analyse the expected number of times agents pull suboptimal arm $k$ as follows,

$$\sum_{\mathcal{C} \in \mathcal{C}} \sum_{i \in \mathcal{C}} \sum_{t=1}^{T} \mathbf{1}\{A_i(t) = k\} \tag{119}$$

$$= \sum_{\mathcal{C} \in \mathcal{C}} \sum_{i \in \mathcal{C}} \sum_{t=1}^{\tau_{k,\mathcal{C}}} \mathbf{1}\{A_i(t) = k\} + \sum_{\mathcal{C} \in \mathcal{C}} \sum_{i \in \mathcal{C}} \sum_{t > \tau_{k,\mathcal{C}}}^{T} \mathbf{1}\{A_i(t) = k\} \tag{120}$$

$$\leq \sum_{\mathcal{C} \in \mathcal{C}} \left( \eta_k + \sum_{i \in \mathcal{C}} D_i(\tau_{k,\mathcal{C}}) + 2|\mathcal{C}| \right) + \sum_{\mathcal{C} \in \mathcal{C}} \sum_{i \in \mathcal{C}} \sum_{t > \tau_{k,\mathcal{C}}}^{T-1} \mathbf{1}\{A_i(t+1) = k\}\mathbf{1}\{N_k^i(t) > \eta_k\}. \tag{121}$$

Taking expectation we have

$$\sum_{\mathcal{C} \in \mathcal{C}_\gamma} \sum_{i \in \mathcal{C}} \sum_{t=1}^{T} \mathsf{P}\left( A_i(t) = k \right) \tag{122}$$

$$\leq \bar{\chi}(G_\gamma)\eta_k + \mathbb{E}\left[ \max_{t \in [T]} \sum_{i=1}^{N} D_i(t) \right] + 2N + \sum_{i=1}^{N} \sum_{t=1}^{T-1} \mathsf{P}\left( A_i(t+1) = k, N_k^i(t) > \eta_k \right) \tag{123}$$

The proof of Lemma 7 follows from Lemma 1 and (123). $\qquad \square$

We upper bound the expected number of outstanding messages by any agent using results by [18] as follows.

**Lemma 8.** *. Let $D_{total}$ be the maximum number of outstanding messages by all the agent at any time step $t \in [T]$ and let $\mathbb{E}[\tau]$ be the expected delay of any message. Then with probability at least $1 - \frac{1}{T}$ we have*

$$\mathbb{E}[D_{total}] \leq N\mathbb{E}[\tau] + 2\log T + 2\sqrt{N\mathbb{E}[\tau]\log T}. \tag{124}$$

*Proof.* The proof directly follows from Lemma 2 by [18]. $\qquad \square$

From Lemmas 7, 3, 4 and 8 we obtain with probability at least $1 - \frac{1}{T}$

$$\text{Reg}_G(T) \leq 8(\xi+1)\sigma_k^2\bar{\chi}(G)\left(\sum_{k>1}\frac{\log T}{\Delta_k}\right) \tag{125}$$

$$+\left(N\mathbb{E}[\tau] + 2\log T + 2\sqrt{N\mathbb{E}[\tau]\log T}\right)\sum_{k>1}\Delta_k \tag{126}$$

$$+ 5N\sum_{k>1}\Delta_k + 4\sum_{i=1}^{N}\left(3\log(3(d_i(G)+1)) + (\log(d_i(G)+1))\right)\sum_{k>1}\Delta_k \tag{127}$$

## D Proof of Theorem 4

We first restate the result for clarity.

**Theorem 7.** *Algorithm 1 obtains, with probability at least $1 - \delta$, cumulative group regret of*

$$\text{Reg}_G(T) = \mathcal{O}\left(KTN\gamma\epsilon + \psi(G_\gamma)\sum_{k\neq k^\star}\frac{\log T}{\Delta_k}\log\left(\frac{K\psi(G_\gamma)\log T}{\delta}\right) + N\Delta_k + \frac{N\log(N\gamma\log T)}{\Delta_k}\right).$$

*Proof.* We decompose the regret based on the dominating set and epoch. Let $\mathcal{I} \subseteq \mathcal{V}$ be an dominating set of $G_\gamma$ and $M_i$ be the number of epochs run for the subgraph covered by agent $i$. Observe that the total regret can be written as,

$$\text{Reg}_G(T) = \sum_{i\in\mathcal{I}}\left(\sum_{k=1}^{K}\sum_{t=1}^{T}\Delta_k \cdot \left(\mathsf{P}(A_i(t)=k) + \sum_{j\in\mathcal{N}_i(G_\gamma)}\mathsf{P}(A_j(t)=k)\right)\right). \tag{128}$$

First, observe that $A_j(t) = A_i(t - d(i,j))$ for all $j \in \mathcal{N}_i(G_\gamma)$ and all $t \in [d(i,j),T]$. Rearranging the above, we have,

$$\text{Reg}_G(T) \leqslant \sum_{i\in\mathcal{I}}\left(\sum_{k=1}^{K}\Delta_k \cdot \left(\sum_{t=1}^{T}\mathsf{P}(A_i(t)=k) + \sum_{j\in\mathcal{N}_i(G_\gamma)}\left(\sum_{t=1}^{T-d(i,j)}\mathsf{P}(A_i(t)=k) + d(i,j)\right)\right)\right) \tag{129}$$

$$\leqslant \sum_{i\in\mathcal{I}}\left(\sum_{k=1}^{K}\Delta_k \cdot |\mathcal{N}_i^+(G_\gamma)| \cdot \left(\sum_{t=1}^{T-\gamma}\mathsf{P}(A_i(t)=k) + \gamma\right)\right) \tag{130}$$

$$= \sum_{i\in\mathcal{I}}\left(|\mathcal{N}_i^+(G_\gamma)|\sum_{k=1}^{K}\Delta_k\left(\sum_{t=1}^{T-\gamma}\mathsf{P}(A_i(t)=k)\right)\right) + N\gamma\sum_{k=1}^{K}\Delta_k. \tag{131}$$

$$\tag{132}$$

Now, observe that we run two algorithms in tandem for each subgraph of $G$ induced by $\mathcal{N}_i^+(G_\gamma)$. Let us split the total number of rounds of the game into epochs that run arm elimination and the intermittent periods of running UCB1. We denote the cumulative regret in the $i^{th}$ induced subgraph from rounds $\gamma$ to $T$ as $\text{Reg}_{\mathcal{N}_i^+(G_\gamma)}(T)$, and analyse it separately.

$$\text{Reg}_{\mathcal{N}_i^+(G_\gamma)}(T) \leqslant |\mathcal{N}_i^+(G_\gamma)|\sum_{k=1}^{K}\left(\Delta_k\left(\sum_{t\leq T-\gamma:t\in\mathcal{M}_i}\mathsf{P}(A_i(t)=k) + \sum_{t\leq T-\gamma:t\notin\mathcal{M}_i}\mathsf{P}(A_i(t)=k)\right)\right). \tag{133}$$

Here $\mathcal{M}_i$ denotes the rounds in which arm elimination is played in the agents in the $i^{th}$ induced subgraph. Since each UCB1 period after each epoch is of length $2\gamma$, we have at most $2\gamma M_i$ rounds of isolated UCB1. We analyse the second term in the bound first. By the standard analysis of the UCB1 algorithm [3], we have that the leader agent, i.e. agent $i$, incurs $\mathcal{O}(K\log T/\Delta)$ regret. We therefore have,

$$|\mathcal{N}_i^+(G_\gamma)|\sum_{k=1}^{K}\left(\Delta_k\left(\sum_{t\notin\mathcal{M}_i}\mathsf{P}(A_i(t)=k)\right)\right) \leqslant |\mathcal{N}_i^+(G_\gamma)| \cdot \sum_{k=1}^{K}\left(\left(1 + \frac{\pi^2}{3}\right)\Delta_k + \frac{8\log(2\gamma M_i)}{\Delta_k}\right).$$

Now, we analyse the first term in the regret bound. By Theorem 8, we have that with probability at least $1 - \delta$ simultaneously for each induced subgraph corresponding to agent $i \in \mathcal{I}$,

$$\sum_{k=1}^{K} \left( \Delta_k \left( \sum_{m \in \mathcal{M}_i} \mathbb{E}\left[ n_k^i(m) \right] \right) \right) = \mathcal{O}\left( \gamma\epsilon \cdot KT |\mathcal{N}_i^+(G_\gamma)| + \sum_{k>1} \frac{\log T}{\Delta_k} \log\left( \frac{K\psi(G_\gamma)}{\delta} \log T \right) \right).$$

Summing over each leader agent, we have that with probability at least $1 - \delta$,

$$\sum_{i \in \mathcal{I}} \sum_{k=1}^{K} \left( \Delta_k \left( \sum_{m \in \mathcal{M}_i} \mathbb{E}\left[ n_k^i(m) \right] \right) \right) = \mathcal{O}\left( \gamma\epsilon \cdot KTN + \sum_{k>1} \frac{\log T}{\Delta_k} \log\left( \frac{K\psi(G_\gamma)}{\delta} \log T \right) \right).$$

Next, observe that for all $i$, $|\mathcal{M}_i| \le \log(MT)$ by Lemma 9. Replacing this result in the UCB1 regret for each leader, and summing over all $i \in \mathcal{I}$, we have,

$$\mathsf{Reg}_G(T) = \mathcal{O}\left( \gamma\epsilon \cdot KTN + \sum_{k>1} \psi(G_\gamma) \frac{\log T}{\Delta_k} \log\left( \frac{K\psi(G_\gamma)\log T}{\delta} \right) + N\Delta_k + \frac{N\log(N\gamma\log T)}{\Delta_k} \right).$$

$\square$

**Lemma 9.** *For any leader $i$, let $L^i(m)$ denote the length of the $m^{th}$ epoch of arm elimination. Then, we have that $L^i(m)$ satisfies,*

$$2^{2m-2}\lambda \le L^i(m) \le K2^{2m-2}\lambda.$$

*Furthermore, the number of arm elimination epochs for agent $i$ satisfies $M_i \le \log_2(T - 2\gamma)$.*

*Proof.* The proof closely follows the proof of Lemma 2 in [15]. For any leader $i$, let $\hat{k}$ be the optimal arm under $r^i(m)$, therefore $r_\star^i(m) - r_{\hat{k}}^i(m) \le 0$ and therefore $\Delta_{\hat{k}}^i(m) = 2^{-m}$, and therefore $L^i(m+1) \ge n_{\hat{k}}^i(m+1) = \lambda(\Delta_{\hat{k}}^i(m))^{-2} \ge 2^{2m}\lambda$. Next, observe that $\Delta_k^i(m) \ge 2^{-m}$ for each arm $k$, and therefore $n_k^i(m+1) \le 2^{2m}\lambda$, giving the upper bound.

For the second part, observe that $\sum_{m=1}^{M_i} L^i(m) \le T - 2\gamma M_i \le T - 2\gamma$, and that $L^i(m) \ge \frac{2^{2m-2}\lambda}{|\mathcal{N}_i^+(G_\gamma)|}$. Summing over $m \in [M_i]$ and taking the logarithm provides us with the result. $\square$

**Lemma 10.** *Denote $\mathcal{E}$ to be the event for which,*

$$\left\{ \forall m, i, k, \left| r_k^i(m) - \mu_k \right| \le 2\gamma\epsilon + \frac{\Delta_k^i(m-1)}{16} \bigwedge \sum_{\substack{t \in \mathcal{M}_i(m) \\ j \in \mathcal{N}_i^+(G_\gamma)}} X_k^j(t + d(i,j)) \le 2n_k^i(m) \right\}$$

*Then, we have that $\mathsf{P}(\mathcal{E}) \ge 1 - \delta$.*

*Proof.* Recall that at each step in the epoch, the leader agent picks an arm $k$ with probability $p_k^i(m) = \frac{n_k^i(m)}{L^i(m)}$, and let $X_k^j(t)$ denote whether agent $j$ picks arm $k$ at time $t$. Let $C_{j \to i}(t) = \tilde{r}_{j \to i}(t) - r_j(t)$ denote the corruption in the transmitted reward from agent $j$ when it reaches agent $i$, and $\mathcal{M}_i(m) = [T_i(m-1) + 1, \cdots, T_i(m)]$ denote the $L^i(m)$ steps in the $m^{th}$ epoch for the arm elimination algorithm run by the leader $i$. We then have,

$$r_k^i(m) = \frac{1}{n_k^i(m)} \left( \sum_{\substack{t \in \mathcal{M}_i(m) \\ j \in \mathcal{N}_i^+(G_\gamma)}} X_k^j(t + d(i,j)) \cdot (r_j(t + d(i,j)) + C_{j \to i}(t + d(i,j))) \right)$$

For simplicity, let

$$A_k^i(m) = \sum_{\substack{t \in \mathcal{M}_i(m) \\ j \in \mathcal{N}_i^+(G_\gamma)}} X_k^j(t+d(i,j)) \cdot r_j(t+d(i,j)), \quad B_k^i(m) = \sum_{\substack{t \in \mathcal{M}_i(m) \\ j \in \mathcal{N}_i^+(G_\gamma)}} X_k^j(t+d(i,j)) \cdot C_{j \to i}(t+d(i,j)).$$

We can bound the first summation by a multiplicative version of the Chernoff-Hoeffding bound [2] as each $r_j$ is bounded within $[0, 1]$ and $X_k^i$ is a random variable in $\{0, 1\}$ with mean $p_k^i(m)L^i(m)\mu_k \leq n_k^i(m)$. We obtain that with probability at least $1 - \beta/2$,

$$\left| \frac{A_k^i(m)}{n_k^i(m)} - \mu_i \right| \leq \sqrt{\frac{3 \log(\frac{4}{\beta})}{n_k^i(m)}}.$$

To bound the second term, we must construct a filtration that ensures that the corruption is measurable. For the set $\mathcal{N}_i^+(G_\gamma)$, consider an order $\sigma$ of the $N$ agents, such that $\sigma[1] = i$, followed by the agents at distance 1 from $i$, then the agents at distance 2, and so on until distance $\gamma$, and next consider the ordering $\{\tilde{r}_\tau\}_{\tau=1}^{|\mathcal{N}_i^+(G_\gamma)|t}$ of the rewards generated by all agents within $\mathcal{M}_i(m)$ where $\tilde{r}_\tau$ is the reward obtained by agent $j = (\sigma(\tau) \mod |\mathcal{N}_i^+(G_\gamma)|)$ during the round $\lfloor \frac{\tau}{|\mathcal{N}_i^+(G_\gamma)|} \rfloor + d(i, j)$, and similarly consider an identical ordering of the pulled arms $\{\tilde{X}_\tau\}_{\tau=1}^{|\mathcal{N}_i^+(G_\gamma)|t}$. Now consider the filtration $\{\mathcal{F}_t\}_{t=1}^{T|\mathcal{N}_i^+(G_\gamma)|}$ generated by the two stochastic processes of $\tilde{r}$ and $\tilde{X}$. Clearly, the corruption $C_{\sigma(j) \to i}(t)$ is deterministic conditioned on $\mathcal{F}_{t-1}$. Moreover, we have that the pulled arm satisfies, for all $\tau \in [|\mathcal{N}_i^+(G_\gamma)|t]$ that $\mathbb{E}[\tilde{X}_\tau | \mathcal{F}_{\tau-1}] = p_k^i(m)$. Furthermore, since the corruption in each round is bounded and deterministic, we have that the sequence $Z_\tau = (\tilde{X}_\tau - p_k^i(m)) \cdot \tilde{C}_\tau$ (where $\tilde{C}_\tau$ is the corresponding ordering of corruptions) is a martingale difference sequence with respect to $\{\mathcal{F}_\tau\}_{\tau=1}^T$. Now, consider the slice of $[|\mathcal{N}_i^+(G_\gamma)|t]$ that is present within $B_k^i(m)$, and let the corresponding indices be given by the set $\widetilde{\mathcal{M}}_i(m)$. Using the fact that the observed rewards are bounded, we have that,

$$\sum_{\tau \in \widetilde{\mathcal{M}}_i(m)} \mathbb{E}[Z_\tau^2 | \mathcal{F}_{\tau-1}] \leq \sum_{\tau \in \widetilde{\mathcal{M}}_i(m)} |\tilde{C}_\tau| \cdot \mathbb{V}(Z_\tau) \leq p_k^i(m) \cdot \sum_{\tau \in \widetilde{\mathcal{M}}_i(m)} \tilde{C}_\tau \leq \gamma C L^i(m).$$

We then have by Freedman's inequality that with probability at least $1 - \frac{\beta}{4}$,

$$\frac{B_k^i(m)}{n_k^i(m)} \leq \frac{p_k^i(m)}{n_k^i(m)} \left( \sum_{\tau \in \widetilde{\mathcal{M}}_i(m)} \tilde{C}_\tau + \frac{\gamma C L^i(m) + \log(4/\beta)}{n_k^i(m)} \right) \leq 2\gamma\epsilon + \sqrt{\frac{\log(4/\beta)}{16 n_k^i(m)}}.$$

The last inequality follows from the fact that $n_k^i(m) \geq \lambda \geq 16 \ln(4/\beta)$. With the same probability, we can derive a bound for the other tail. Now, observe that since each $X_k^i$ is a random variable with mean $p_k^i$, we have by the multiplicative Chernoff-Hoeffding bound that the probability that the sum of $L^i(m)$ i.i.d. bernoulli trials with mean $p_k^i(m)$ is greater than $2p_k^i(m) \cdot L^i(m) = 2n_k^i(m)$ is at most $2 \exp(-n_k^i(m)/3) \leq 2 \exp(-\lambda/3) \leq \beta$.

To conclude the proof, we apply each of the above bounds with $\beta = \frac{\delta}{2K\alpha(G_\gamma) \log T}$ to each epoch and arm. Observe that $\beta \geq 4 \exp\left(-\frac{\lambda}{16}\right)$. Now, since $\log(4/\beta) = \lambda/(32)^2$ we have that,

$$\mathbb{P}\left( |r_k^i(m) - \mu_k| \geq 2\gamma\epsilon + \frac{\Delta_k^i(m-1)}{16} \bigwedge \sum_{\substack{t \in \mathcal{M}_i(m) \\ j \in \mathcal{N}_i^+(G_\gamma)}} X_k^j(t + d(i, j)) \geq 2n_k^i(m) \right) \leq \frac{\delta}{2K\alpha(G_\gamma) \log T}.$$

The proof concludes by a union bound over all epochs, arms and agents in $\mathcal{I}$. $\qquad \square$

**Lemma 11.** *If the event $\mathcal{E}$ (Lemma 10) occurs then for each $i \in \mathcal{I}, m \in \mathcal{M}_i$,*

$$-2\gamma\epsilon - \frac{\Delta_\star^i(m-1)}{8} \leq r_\star^i(m) - \mu_\star \leq 2\gamma\epsilon.$$

*Proof.* Observe that $r_\star^i(m) \geq r_{k^\star}^i(m) - \frac{1}{16}\Delta_{k^\star}^i(m-1)$. This fact coupled with the fact that $\mathcal{E}$ holds provides the lower bound. The upper bound is obtained by observing that,

$$r_\star^i(m) \leq \max_i \left\{ \mu_i + 2\gamma\epsilon + \frac{\Delta_k^i(m-1)}{16} - \frac{\Delta_k^i(m-1)}{16} \right\} \leq \mu_\star + 2\gamma\epsilon.$$

$\qquad \square$

**Lemma 12.** *If the event $\mathcal{E}$ (Lemma 10) occurs then for each $i \in \mathcal{I}, m \in \mathcal{M}_i$,*

$$\Delta_k^i(m) \geq \frac{\Delta_k}{2} - 6\gamma\epsilon \sum_{n=1}^{m} 8^{n-m} - \frac{3}{4}2^{-m}.$$

*Proof.* We first bound $\Delta_k^i(m) \leq 2(\Delta_k + 2^{-m} + 2\gamma\epsilon \cdot \sum_{n=1}^{m} 8^{n-m})$ under $\mathcal{E}$ by induction. Observe that when $m = 1$ we have that trivially $\Delta_k^i(1) \leq 1 \leq 2 \cdot 2^{-1}$. Now, if the bound holds for epoch $m - 1$ for any agent, we have by Lemma 11,

$$r_\star^i(m) - r_k^i(m) = r_\star^i(m) - \mu_\star + \mu_\star - \mu_k + \mu_k - r_k^i(m) \leq 4\gamma\epsilon + \Delta_k + \frac{\Delta_k^i(m-1)}{16}.$$

Replacing the induction hypothesis in the upper bound, we have,

$$r_\star^i(m) - r_k^i(m) \leq 4\gamma\epsilon + \Delta_k + \frac{1}{8}\left(\Delta_k + 2^{-(m-1)} + 2\gamma\epsilon \cdot \sum_{n=1}^{m-1} 8^{n-m+1}\right)$$

$$\leq 2(\Delta_k + 2^{-m} + 2\gamma\epsilon \cdot \sum_{n=1}^{m} 8^{n-m}).$$

Now, we bound the gaps as,

$$\Delta_k^i(m) \geq r_\star^i(m) - r_k^i(m) \geq \Delta_k - 4\gamma\epsilon - \left(\frac{\Delta_{k^\star}^i(m-1)}{8} - \frac{\Delta_k^i(m-1)}{16}\right).$$

The last inequality follows from Lemma 11 and the event $\mathcal{E}$. Replacing the bound from induction we obtain,

$$\Delta_k^i(m) \geq \Delta_k - 4\gamma\epsilon - \left(\frac{6\gamma\epsilon}{8}\sum_{n=1}^{m} 2^{n-m} + \frac{3}{8}2^{-(m-1)} + \frac{\Delta_k}{8}\right)$$

$$\geq \frac{\Delta_k}{2} - 6\gamma\epsilon \sum_{n=1}^{m} 8^{n-m} - \frac{3}{4}2^{-m}.$$

$\square$

**Theorem 8.** *The cumulative regret for all agents within each independent set corresponding to leader $i \in \mathcal{I}$ satisfy simultaneously, with probability at least $1 - \delta$,*

$$\sum_{m=1}^{\mathcal{M}_i}\sum_{k=1}^{K} \Delta_k \mathbb{E}[n_k^i(m)] = \mathcal{O}\left(\log\left(\frac{K\psi(G_\gamma)}{\delta}\log(T)\right)\log(T)\left(\sum_{k=1}^{K}\frac{1}{\Delta_k}\right) + \gamma\epsilon \cdot KT \cdot |\mathcal{N}_i^+(G_\gamma)|\right).$$

*Proof.* We bound the regret in each epoch $m \in \mathcal{M}_i$ for each arm $k \neq k^\star$ based on three cases.

**Case 1.** $0 \leq \Delta_k \leq 4/2^m$: We have that $n_k^i(m) \leq \lambda 2^{2(m-1)}$ since $\Delta_k^i(m-1) \geq 2^{m-1}$, and hence,

$$\Delta_k \mathbb{E}[n_k^i(m)] \leq \frac{4\lambda}{\Delta_k^2} \cdot \Delta_k = 4\lambda \cdot \frac{1}{\Delta_k}.$$

**Case 2.** $\Delta_k > 4/2^m$ and $\gamma\epsilon \sum_{n=1}^{m} 8^{n-m} \leq \Delta_k/64$: We have by Lemma 12,

$$\Delta_k^i(m) \geq \frac{\Delta_k}{2} - 6\gamma\epsilon \sum_{n=1}^{m} 8^{n-m} - \frac{3}{4}2^{-m} \geq \Delta_k\left(\frac{1}{2} - \frac{3}{32} - \frac{3}{8}\right) = \frac{\Delta_k}{32}.$$

Therefore, we have that $n_k^i(m) \leq \frac{1024\lambda}{\Delta_k^2}$, and hence the regret is,

$$\Delta_k \mathbb{E}[n_k^i(m)] \leq \frac{1024\lambda}{\Delta_k^2} \cdot \Delta_k = 1024\lambda \cdot \frac{1}{\Delta_k}.$$

**Case 3.** $\Delta_k > 4/2^m$ and $\gamma\epsilon \sum_{n=1}^m 8^{n-m} > \Delta_k/64$: This implies that $\Delta_k \leq 64\gamma\epsilon \cdot \sum_{n=1}^m 8^{n-m}$. Therefore,

$$\Delta_k \mathbb{E}[n_k^i(m)] \leq 64\lambda\gamma\epsilon \left(\sum_{n=1}^m 8^{n-m}\right) \cdot 2^{2(m-1)}$$

$$\leq 64\lambda\gamma\epsilon \left(\frac{8^{m+1}}{7}\right) \cdot \frac{2^{2(m-1)}}{2^{3m}}$$

$$\leq \frac{512}{7}\gamma\epsilon \cdot L^i(m).$$

Here the last inequality follows from Lemma 9. Putting it together and summing over all epochs and arms, we have with probability at least $1 - \delta$ simultaneously for each $i \in \mathcal{I}$,

$$\sum_{m=1}^{\mathcal{M}_i} \sum_{k=1}^K \Delta_k \mathbb{E}[n_k^i(m)] \leq 1024^2 \log\left(\frac{8K\psi(G_\gamma)}{\delta} \log(T)\right) \log(T) \left(\sum_{k=1}^K \frac{1}{\Delta_k}\right) + 74\gamma\epsilon \cdot KT \cdot |\mathcal{N}_i^+(G_\gamma)|.$$

$\square$

# E    Proof of Theorem 5

In this section we consider that each agent passes messages upto $\gamma$-hop neighbors. Agents do not use the messages received during last $\bar{\gamma}$ number of time steps.

**Lemma 13.** *Let $\bar{\chi}(G_\gamma)$ is the clique number of graph $G_\gamma$. Let $\eta_k = \left(\frac{8(\xi+1)\sigma_k^2}{\Delta_k^2}\right) \log T$. Then we have*

$$\sum_{i=1}^N \mathbb{E}[n_k^i(T)] \leq \bar{\chi}(G_\gamma)\eta_k + (N - \bar{\chi}(G_\gamma))(\bar{\gamma} + \gamma - 1) + 2N + \tag{134}$$

$$+ \sum_{i=1}^N \sum_{t=1}^{T-1} \left[\mathsf{P}\left(\widehat{\mu}_1^i(t) \leq \mu_1 - C_1^i(t)\right) + \mathsf{P}\left(\widehat{\mu}_k^i(t) \geq \mu_k + C_k^i(t)\right)\right] \tag{135}$$

*Proof.* Let $\mathcal{C}_\gamma$ be a non overlapping clique covering of $G_\gamma$. Note that for each suboptimal arm $k > 1$ we have

$$\sum_{i=1}^N \mathbb{E}[n_k^i(T)] = \sum_{i=1}^N \sum_{t=1}^T \mathsf{P}\left(A_i(t) = k\right) = \sum_{\mathcal{C}\in\mathcal{C}_\gamma} \sum_{i\in\mathcal{C}} \sum_{t=1}^T \mathsf{P}\left(A_i(t) = k\right). \tag{136}$$

Let $\tau_{k,\mathcal{C}}$ denote the maximum time step when the total number of times arm $k$ has been played by all the agents in clique $\mathcal{C}$ is at most $\eta_k + (|\mathcal{C}| - 1)(\bar{\gamma} + \gamma - 1) + |\mathcal{C}|$ times. This can be stated as $\tau_{k,\mathcal{C}} := \max\{t \in [T] : \sum_{i\in\mathcal{C}} n_k^i(t) \leq \eta_k + (|\mathcal{C}| - 1)(\bar{\gamma} + \gamma - 1) + |\mathcal{C}|\}$. Then, we have that $\eta_k + (|\mathcal{C}| - 1)(\bar{\gamma} + \gamma - 1) < \sum_{i\in\mathcal{C}} n_k^i(\tau_{k,\mathcal{C}}) \leq \eta_k + (\mathcal{C} - 1)(\bar{\gamma} + \gamma - 1) + |\mathcal{C}|$.

For each agent $i \in \mathcal{C}$ let

$$\bar{N}_k^i(t) := \sum_{\tau=1}^t \mathbf{1}\{A_i(\tau) = k\} + \sum_{j\neq i,j\in\mathcal{C}} \sum_{\tau=1}^{t-\bar{\gamma}} \sum_{\tau'=1}^\tau \mathbf{1}\{A_j(\tau') = k\}\mathbf{1}\{(i,j) \in E_{\tau',\tau}\},$$

denote the sum of the total number of times agent $i$ pulled arm $k$ and the total number of observations it received from agents in its clique about arm $k$ until time $t$.

Note that for all $i \in \mathcal{C}$ we have $N_k^i(t) > \eta_k, \forall t > \tau_{k,\mathcal{C}}$.

We analyse the expected number of times agents pull suboptimal arm $k$ as follows,

$$\sum_{\mathcal{C}\in\mathcal{C}_\gamma} \sum_{i\in\mathcal{C}} \sum_{t=1}^T \mathbf{1}\{A_i(t) = k\} \tag{137}$$

$$= \sum_{\mathcal{C} \in \mathcal{C}_\gamma} \sum_{i \in \mathcal{C}} \sum_{t=1}^{\tau_{k,\mathcal{C}}} \mathbf{1}\{A_i(t) = k\} + \sum_{\mathcal{C} \in \mathcal{C}_\gamma} \sum_{i \in \mathcal{C}} \sum_{t > \bar{\tau}^i_{k,\mathcal{C}}}^{T} \mathbf{1}\{A_i(t) = k\} \tag{138}$$

$$\leq \sum_{\mathcal{C} \in \mathcal{C}_\gamma} (\eta_k + (|\mathcal{C}| - 1)(\bar{\gamma} + \gamma - 1) + 2|\mathcal{C}|) + \sum_{\mathcal{C} \in \mathcal{C}_\gamma} \sum_{i \in \mathcal{C}} \sum_{t > \tau_{k,\mathcal{C}}}^{T-1} \mathbf{1}\{A_i(t+1) = k\}\mathbf{1}\{N_k^i(t) > \eta_k\}. \tag{139}$$

Taking expectation we have

$$\sum_{\mathcal{C} \in \mathcal{C}_\gamma} \sum_{i \in \mathcal{C}} \sum_{t=1}^{T} \mathsf{P}\left(A_i(t) = k\right) \tag{140}$$

$$\leq \sum_{\mathcal{C} \in \mathcal{C}_\gamma} (\eta_k + (|\mathcal{C}| - 1)(\bar{\gamma} + \gamma - 1) + 2|\mathcal{C}|) + \sum_{\mathcal{C} \in \mathcal{C}_\gamma} \sum_{i \in \mathcal{C}} \sum_{t > \tau_{k,\mathcal{C}}}^{T-1} \mathsf{P}\left(A_i(t+1) = k, N_k^i(t) > \eta_k\right). \tag{141}$$

$$= \bar{\chi}(G_\gamma)\eta_k + (N - \bar{\chi}(G_\gamma))\left(\bar{\gamma} + \gamma - 1\right) + 2N + \sum_{\mathcal{C} \in \mathcal{C}_\gamma} \sum_{i \in \mathcal{C}} \sum_{t=1}^{T-1} \mathsf{P}\left(A_i(t+1) = k, N_k^i(t) > \eta_k\right) \tag{142}$$

The proof of Lemma 13 follows from Lemma 1 and (142). $\qquad \square$

Now we prove Theorem 5 as follows. Thus using Lemmas 4, 5 and 13 we obtain

$$\mathsf{Reg}_G(T) \leq 8(\xi + 1)\sigma_k^2 \bar{\chi}(G_\gamma) \left( \sum_{k>1} \frac{\log T}{\Delta_k} \right) + ((N - \bar{\chi}(G_\gamma)(\bar{\gamma} + \gamma - 1) + 5N) \sum_{k>1} \Delta_k \tag{143}$$

$$+ 4 \sum_{i=1}^{N} \left(3 \log(3(d_i(G_\gamma) + 1)) + (\log\left(d_i(G_\gamma) + 1\right))\right) \sum_{k>1} \Delta_k \tag{144}$$

# F  Lower Bounds

**Theorem 9** (Minimax Rate). *For any multi-agent algorithm $\mathcal{A}$, there exists a $K-$armed environment over $N$ agents with $\Delta_k \leq 1$ such that,*

$$\mathsf{Reg}_G(\mathcal{A}, T) \geqslant c\sqrt{KN(T + \widetilde{d}(G))}.$$

*Furthermore, if $\mathcal{A}$ is an agnostic decentralized policy, there exists a $K-\mathrm{armed}$ environment over $N$ agents with $\Delta_k \leq 1$ for any connected graph $G$ and $\gamma \geq 1$ such that, for some absolute constant $c'$*

$$\mathsf{Reg}_G(\mathcal{A}, T) \geqslant c'\sqrt{\alpha^\star(G_\gamma)KNT}.$$

*Where $\tilde{d}(G) = \sum_{i=1}^{d^\star(G)} \bar{d}_{=i} \cdot i$ denotes the average delay incurred by message-passing across the network $G$, $d_{=i} = \frac{1}{N} \sum_{i,j} \mathbb{1}\{d(i,j) = i\}$ denotes the number of agent pairs that are at distance exactly $i$, and $\alpha^\star(G_\gamma) = \frac{N}{1+\bar{d}_\gamma}$ is Turan's lower bound [32] on $\alpha(G_\gamma)$.*

*Proof.* Our approach is an extension of the single-agent bandit lower bound [6]. Let $\mathcal{A}$ be a deterministic (multi-agent) algorithm, and let the empirical distribution of arm pulls across all agents be given by $p^i(t) = \left(p_1^i(t), ..., p_K^i(t)\right)$, where $p_k(t) = \frac{n_k^i(T)}{T}$. Consider the random variable $J_t^i$ drawn according to $p^i(t)$ and $\mathsf{P}_i$ denote the law of $J_t$ when drawn from arm $k$ having parameter $\frac{1+\varepsilon}{2}$ (and other arms with parameter $\frac{1-\varepsilon}{2}$). We have,

$$\mathsf{P}_k\left(J_t^i = j\right) = \mathbb{E}_k\left[\frac{n_i^k(T)}{T}\right].$$

Since on pulling any arm $k' \neq k$, we obtain regret $\varepsilon$, we therefore have for the group regret,

$$\mathbb{E}_k \left[ \sum_{t=1}^{T} \left( N \cdot r_k(t) - \sum_{i \in \mathcal{V}} r_{A_i}(t) \right) \right] = \varepsilon \cdot T \cdot \sum_{i \in \mathcal{V}} \mathsf{P}_k \left( J_t^i = k' \right)$$

$$= \varepsilon \cdot T \cdot \sum_{i \in \mathcal{V}} \left( 1 - \sum_{k' \neq k} \mathsf{P}_k \left( J_t^i = k' \right) \right).$$

By Pinsker's inequality and averaging over all $k \in [K]$, we have for any $i \in \mathcal{V}$,

$$\frac{1}{K} \sum_{k=1}^{K} \mathsf{P}_k \left( J_t^i = k \right) \leqslant \frac{1}{K} + \frac{1}{K} \sum_{k=1}^{K} \sqrt{\frac{1}{2} \mathsf{KL}(\mathsf{P}_0, \mathsf{P}_k)}.$$

We now bound the R.H.S. using the chain rule for KL-divergence. Since we assume that $\mathcal{A}$ is deterministic, we have that the rewards obtained by the agent $i$ until time $t$ from its neighborhood alone determine uniquely the empirical distribution of plays. Here, the analysis diverges from that of the single-agent bandit as a richer set of observations is available to each agent. Denote the set of rewards observed by agent $i$ at instant $\tau$ be given by $\mathcal{O}_i(\tau)$. First, observe that since each reward is i.i.d., we have for any $k$,

$$\mathsf{KL}(\mathsf{P}_0(\mathcal{O}_i(\tau)), \mathsf{P}_k(\mathcal{O}_i(\tau))) = |\mathcal{O}_i(\tau)| \cdot \mathsf{KL} \left( \frac{1-\varepsilon}{2}, \frac{1+\varepsilon}{2} \right)$$

For $k = 0$ the above divergence is 0. When we consider the standard single-agent setting, $|\mathcal{O}_i(\tau)| = 1$, recovering the usual bound. Now, by the chain rule, we have that, at round $t$ for any agent $i$, and arm $k \in [K]$,

$$\mathsf{KL}(\mathsf{P}_0(t), \mathsf{P}_k(t)) = \mathsf{KL}(\mathsf{P}_0(1), \mathsf{P}_k(1)) + \sum_{\tau=2}^{t} |\mathcal{O}_i(\tau)| \, \mathsf{KL} \left( \frac{1-\varepsilon}{2}, \frac{1+\varepsilon}{2} \right)$$

$$= \mathsf{KL} \left( \frac{1-\varepsilon}{2}, \frac{1+\varepsilon}{2} \right) \mathbb{E}_0 \left[ \sum_{j \in \mathcal{V}} n_j^k(t - d(i,j)) \right].$$

Replacing this result in the earlier equation, we have by the concavity of KL divergence:

$$\frac{1}{K} \sum_{k=1}^{K} \mathsf{P}_k \left( J_t^i = k \right) \leqslant \frac{1}{K} + \frac{1}{K} \sum_{k=1}^{K} \sqrt{\frac{1}{2} \mathsf{KL}(\mathsf{P}_0, \mathsf{P}_k)}$$

$$\leqslant \frac{1}{K} + \frac{1}{K} \sum_{k=1}^{K} \sqrt{\mathsf{KL} \left( \frac{1-\varepsilon}{2}, \frac{1+\varepsilon}{2} \right) \mathbb{E}_0 \left[ \sum_{j \in \mathcal{V}} n_j^k(T - d(i,j)) \right]}$$

$$\leqslant \frac{1}{K} + \sqrt{\left( \frac{TN - \sum_{j=1}^{d^\star(G)} d_{=j}(i) \cdot j}{K} \right) \cdot \mathsf{KL} \left( \frac{1-\varepsilon}{2}, \frac{1+\varepsilon}{2} \right)}.$$

Now, observe that the KL divergence between Bernoulli bandits can be bounded as

$$\mathsf{KL}(p, q) \leq \frac{(p-q)^2}{q(1-q)}.$$

Substituting we get,

$$\frac{1}{K} \sum_{k=1}^{K} \mathsf{P}_k \left( J_t^i = k \right) \leqslant \frac{1}{K} + \sqrt{\frac{4\varepsilon^2 (NT - \sum_{j=1}^{d^\star(G)} d_{=j}(i) \cdot j)}{(1-\varepsilon^2)K}}.$$

Replacing this in the regret and using $\varepsilon \leqslant 1/2$, we get that,

$$\mathbb{E}_k \left[ \sum_{t=1}^{T} \left( N \cdot r_k(t) - \sum_{i \in \mathcal{V}} r_{A_i}(t) \right) \right]$$

$$\geqslant \varepsilon \cdot T \cdot \sum_{i \in \mathcal{V}} \left( 1 - \frac{1}{K} - \sqrt{\frac{4\varepsilon^2 (NT - \sum_{j=1}^{d^\star(G)} d_{=j}(i) \cdot j)}{(1-\varepsilon^2)K}} \right)$$

$$\geqslant \varepsilon \cdot T \cdot \sum_{i \in \mathcal{V}} \left( \frac{1}{2} - 4\varepsilon \sqrt{\frac{(NT - \sum_{j=1}^{d^\star(G)} d_{=j}(i) \cdot j)}{3K}} \right)$$

$$= \frac{\varepsilon \cdot NT}{2} - \frac{4\varepsilon^2 NT}{\sqrt{K}} \left( \sum_{i,j \in \mathcal{V}} T - d(i,j) \right)^{1/2}$$

Setting $\varepsilon = c \cdot \sqrt{\frac{K}{N(T - \sum_{j=1}^{d^\star(G)} \bar{d}_{=j} \cdot j)}}$ where $c$ is a constant to be tuned later, we have,

$$\mathbb{E}_k \left[ \sum_{\tau=1}^{T} \left( N \cdot r_{k,t} - \sum_{i \in \mathcal{V}} r_{A_i(t),t} \right) \right] \geqslant \left( \frac{c}{2} - \frac{4c^2}{\sqrt{3}} \right) \cdot \sqrt{\frac{KN^2T^2}{N(T - \sum_{j=1}^{d^\star(G)} \bar{d}_{=j} \cdot j)}}$$

$$\geqslant 0.027 \sqrt{KN(T + \sum_{j=1}^{d^\star(G)} \bar{d}_{=j} \cdot j)}.$$

This proves the first part of the theorem. Now, when the policies are decentralized and agnostic, the chain rule step can be factored as follows.

$$\mathsf{KL}(\mathsf{P}_0(t), \mathsf{P}_k(t)) = \mathsf{KL}(\mathsf{P}_0(1), \mathsf{P}_k(1)) + \sum_{\tau=2}^{t} |\mathcal{O}_i(\tau)| \, \mathsf{KL}\left( \frac{1-\varepsilon}{2}, \frac{1+\varepsilon}{2} \right)$$

$$= \mathsf{KL}\left( \frac{1-\varepsilon}{2}, \frac{1+\varepsilon}{2} \right) \mathbb{E}_0 \left[ \sum_{j \in \mathcal{N}_\gamma^+(G)} n_j^k(t - d(i,j)) \right].$$

Note that here instead of taking the cumulative sum over all $\mathcal{V}$ we select only those agents that are within the $\gamma-$neighborhood of $i$ in $G$, since conditioned on these observations the rewards of the agents are independent of all other rewards (by Assumption), and hence the higher-order KL divergence terms are 0. Replacing this in the analysis gives us the following decomposition (after similar steps as the first part):

$$\mathbb{E}_k \left[ \sum_{t=1}^{T} \left( N r_k(t) - \sum_{i \in \mathcal{V}} r_{A_i}(t) \right) \right] \geqslant \frac{NT\varepsilon}{2} - \frac{4\varepsilon^2 T}{\sqrt{3K}} \cdot \sum_{i \in \mathcal{V}} \left( \sum_{j:\mathcal{N}_\gamma^+(i)} T - d(i,j) \right)^{1/2}$$

$$\geqslant \frac{NT\varepsilon}{2} - \frac{4\varepsilon^2 N^{1/2} T}{\sqrt{3K}} \cdot \left( \sum_{i \in \mathcal{V}} \sum_{j:\mathcal{N}_\gamma^+(i)} T - d(i,j) \right)^{1/2}$$

Setting $\varepsilon = c \cdot \sqrt{\frac{NK}{\sum_{i \in \mathcal{V}} \sum_{j:\mathcal{N}_\gamma^+(i)} T - d(i,j)}}$ where $c$ is a constant to be tuned later, we have,

$$\mathbb{E}_k \left[ \sum_{t=1}^{T} \left( N \cdot r_k(t) - \sum_{i \in \mathcal{V}} r_{A_i}(t) \right) \right] \geqslant \left( \frac{c}{2} - \frac{4c^2}{\sqrt{3}} \right) \cdot \sqrt{\frac{N^3 T^2}{\sum_{i \in \mathcal{V}} \sum_{j \in \mathcal{N}_i^+(G_\gamma)} T - d(i,j)}}$$

$$\geqslant \left( \frac{c}{2} - \frac{4c^2}{\sqrt{3}} \right) \cdot \sqrt{\frac{N^3 T}{\sum_{i \in \mathcal{V}} 1 + d_i(G_\gamma)}}$$

$$\geqslant \frac{3}{4} \left( \frac{c}{2} - \frac{4c^2}{\sqrt{3}} \right) \sqrt{\alpha^\star(G_\gamma)NT}$$

$$\geqslant 0.019 \sqrt{\alpha^\star(G_\gamma)NT}.$$

The constants in both settings are obtained by optimizing $c$ over $\mathbb{R}$. Extending this to random (instead of deterministic) algorithms is straightforward via Fubini's theorem, see Theorem 2.6 of Bubeck [5]. □

# G  Pseudo code

---

**Algorithm 2:** RCL-LF

---

**Input:** Arms $k \in [K]$, variance proxy upper bound $\sigma^2$, parameter $\xi$
**Initialize:** $N_k^i(0) = \widehat{\mu}_k^i(0) = C_k^i(0) = 0, \forall k, i$
**for** each iteration $t \in [T]$ **do**
    **for** each agent $i \in [N]$ **do**
        /\* Sampling phase                                                 \*/
        **if** $t = 1$ **then**
            $A_t^i \leftarrow \text{RANDOMARM}\,([K])$
        **end**
        **else**
            $A_t^i \leftarrow \arg\max_k \widehat{\mu}_k^i(t-1) + C_k^i(t-1)$
        **end**
        /\* Send messages                                                   \*/
        $\text{CREATE}\left(\mathbf{m}_t^i := \left\langle A_t^i, r_t^i, i, t \right\rangle\right)$
        $\text{SEND}\left(\mathbf{M}_t^i \leftarrow \mathbf{M}_{t-1}^i \cup \mathbf{m}_t^i\right)$
    **end**
    **for** each agent $i \in [N]$ **do**
        /\* Receive messages                                            \*/
        **for** each neighbor $j \in \mathcal{N}_i(G_\gamma)$ **do**
            /\* Discard messages with probability $1 - p_i$                 \*/
            **for** each message $\mathbf{m} \in \mathbf{M}_t^j$ **do**
                with probability $p_i$,    $\mathbf{M}_t^i \leftarrow \mathbf{M}_t^i \cup \mathbf{m}$
                with probability $1 - p_i$,    $\mathbf{M}_t^i \leftarrow \mathbf{M}_t^i$
            **end**
        **end**
        /\* Update estimates                                            \*/
        **for** each arm $k \in [K]$ **do**
            $\text{CALCULATE}\left(N_k^i(t), \widehat{\mu}_k^i(t), C_k^i(t)\right)$
        **end**
    **end**
**end**

---