# OpenReview forum: "One More Step Towards Reality: Cooperative Bandits with Imperfect Communication"
_NeurIPS.cc/2021/Conference — NeurIPS 2021 Poster_

### Official Review · Reviewer_9KQ6 · 2021-07-13

**Rating:** 5
**Confidence:** 4

**Summary:**

Cooperative stochastic bandits are studied under several imperfect communications models. Most results study, for each node, the information provided by a clique to which it belongs, resulting in the clique covering number appearing as a factor of log(T) in many bounds. The imperfect communication is also reflected in the regret bounds. There is also an algorithm proposed for perfect communication and a lower bound.

**Limitations And Societal Impact:**

--

**Main Review:**

This paper is a mixture of many simple extensions of other analyses. For instance, some results just consist on quantify how much delay one incurs and then one just need to apply well-known results about regret of stochastic bandits under delays. Still, the results provide some value. Although some choices taken seem a bit arbitrary.

For example, your results say that you can drop messages and pay some quantifiable regret for it. You argue that dropping messages at some particular rate for each node allows to make things more unbiased, but there is no theoretical nor empirical evidence in this paper for such a claim.  The bounds actually indicate that not discarding messages is best to have a better upper bound on the regret.

These factors you obtain multiplying log T can be quite a high price to pay, can't they? If you don't analyse things in cliques but do over the whole network you might get some extra constant regret in the bound but you will not get these factors in the log T. If you don't have a factors multiplying the log T it means that this part of the regret will not be as large as the corresponding part with the factors until T is exponential in the first case (wrt to some fixed T in the second one)


In thm 4 one needs to have an algorithm that needs to compute a clique covering.

Many lemmas are essentially the same. I feel presentation could be done to compress them into one.

What happens with UCB and delays is already well known (Jouliani et al) even if there are errors in the estimates of the means (Martinez-Rubio et al).

(Cesa-Bianchi and Lugosi, 2006) is cited in the appendix but it does not appear in the list of references.


Minor typos/suggestions:

+ l307 O(\sum_{k>1}...) do you mean \Omega ?

+ Figure 3 there are two "(a)"s and no "(b)"s

+ l343 comma after [N]
    + Also, a space is missing before \forall

+ Algorithm 1:
    + second line. Add commas between different parameters, like before lambda.
    + I don't understand why some m's have _i and some don't. I think all of them should not have the _i
    + Where is the definiton of T_i(m) and \Delta_{k}^i(m) when m > 0?

+ centered equation of statement of lemma 1 (and others). N_k^i should be mathcal{N}_k^i according to the notation table

+ lemma 2. "is the clique number of graph G" -> "be the clique number of graph G"

+ Equation (6). The |\mathcal{C}| could be in the line below to avoid confusion since it really belongs to the third summand.

+ l508 "this results" ->  "these results"

+ In (26), the second instance of the notation for the clique covering number is wrong.

+ l518 "$\forall \tau > 0$." ->  "$\forall \tau > 0$:"

+ l540 (59). You could mention that you first bound the first 3 summands and then the rest is bounded by the integral.

+ (62) I believe there should be a 3 multiplying (d_i+1) inside of the log in the right hand side of (62). So in the statement of the lemma there should be a 3 as well multiplying the (d_i+1) in the second log (equation (58)). The final bound of theorem 1 should also change accordingly

+ l555 "similar to proof of" ->  "similar to the proof of"

+ l555 "We star" ->  "We start"

+ l608 "by all the agent" ->  "by all the agents"

+ l609 "is the clique number" ->  "be the clique number". Same in l721

+ l685 There is a missing citation (I presume it would be to Angluint and Valiant 1979)

+ l707 "satisfy" ->  "satisfies"


**Time Spent Reviewing:**

8

---

> ### Author Response · Authors · 2021-08-11
> **Author Response**
>
> Thank you for the constructive feedback! In the following, we address your concerns individually.
>
> ---**Discarding messages with reducing the information bias across the network:** Due to the complexity of the joint probability of agent-action space it is difficult to provide a tight upper bound for regret of individual agents. Thus it is difficult to provide theoretical guarantees on how information discarding reduces the bias across the network and improves group performance. However, we provide an intuitive explanation and provide empirical results to validate this.
>
> Intuitive explanation:
>
> First, consider the instantaneous reward sharing case. In this case, each agent is sharing the instantaneous action and reward with its neighbors and those messages are not forwarded to other agents. In this case, it is clear that the number of messages (amount of information) received by each agent is proportional to the degree (i.e., number of neighbors) of the agent. As explained in the first two paragraphs of Section 3, this creates an information disparity among agents. Well-connected agents (agents with a high degree) receive more information and perform better than poorly connected agents. As a result of this disparity group performance decreases. The effect of information disparity is exacerbated in the presence of link failures. Thus in order to regulate the information flow in the network in our algorithm each agent, $i$ incorporates received information with a probability inversely proportional to their degree (i.e. p_i=\frac{d_{\min}}{d_{i}}). This is equivalent to discarding messages with $1-p_i$. Thus all the agents approximately receive the same amount of information regardless of their degree (number of neighbors), leading to improved group performance.
>
> Consider the case where agents communicate over a star graph. The center agent has more neighbors and hence receives more information. Peripheral agents have only one neighbor and they receive less information through communication. This leads to center agents performing better and peripheral agents performing poorly. As a result group performance decreases. When agents incorporate received messages with a probability inversely proportional to the degree center agent incorporate less information and peripheral agents incorporate nearly all information. As a result all the agents in the network approximately use the same amount of information. This improves the group performance.
>
> Now consider the message passing case. In this paper, we consider the case where each message is forwarded at most $\gamma$ times through $G$. This is equivalent to the case where each agent can send messages only up to $\gamma$ hop neighbor. When  $\gamma = d_\star(G)$ (diameter of the graph $G$) we have that $G_{\gamma}$ is a complete graph. And we have $p_i=\frac{d_{\min}(G_{\gamma})}{d_i(G_{\gamma})}=1$ and agents do not discard any message. However, when $\gamma < d_\star(G)$ the graph  $G_{\gamma}$ is not complete. Therefore agents receive different amounts of information which are approximately proportional to the degree distribution of $G_{\gamma}.$ As explained in the previous paragraph this information disparity leads to a performance disparity among agents. As a result group performance decreases.  In this case we design the algorithm such that each agent $i$ discards messages with $1-p_i$ where $p_i=\frac{d_{\min}(G_{\gamma})}{d_i(G_{\gamma})}.$ This regulates the information flow mitigating the bias introduced by information disparity. As a result, the group obtains near-optimal performance.
>
> In summary, information disparity created by different agents receiving different amounts of information leads to poor group performance. We use message discarding to reduce the information disparity introduced by the degree distribution of the network and communication parameter $\gamma.$ Hence our algorithms obtain improved performance.
>
> Experimental results:
> Here we consider a simple example with a star network with 10 peripheral agents. We consider 10  arms ($K=10$) with Gaussian reward distributions. Expected reward for the optimal arm  is $\mu_{1}=11$ and for all sub-optimal options $k\geq 2$ is $\mu_{k}=10$. We let variance associated with all arms $k$ be $\sigma_k^2 =1$. Because the expected reward gaps  $\Delta_k=1$, $k\geq 2$, are equal to the variances $\sigma_k^2 = 1$, it is a challenging problem to distinguish the optimal arm from the sub-optimal options. For all simulations, we consider 1000 time steps ($T=1000$) and use 1000 Monte Carlo simulations. We consider communication link failure to be $p=0.8.$ We provide results for the average expected regret of center agents. Average expected regret of peripheral agents and average expected regret of the group at time $t=1000.$
>
> |                        | average center | average peripheral  | average group |
> |------------------------|----------------|---------------------|---------------|
> | $p_i=1,\forall i\in N$ | 17.45          | 57.67             | 54.01       |
> | $p_i=\frac{1}{d_i},\forall i\in N$    | 26.83          | 27.50              | 27.03       |
>
> These results empirically verify our intuition. We see that when we reduce the information disparity group performance significantly improves.
>
> ---**Constant factors that multiply log T:** Analysing the regret of the whole network without considering a clique partitioning leads to poor regret bounds. (large constants multiplying $\log T$ term.) This is due to the fact that messages created by an agent do not propagate to all other agents. Since clique is a complete graph,  messages created by an agent propagate to all other agents in that clique. This allows us to find tight bounds for the regret in each clique.
>
> ---**Algorithm for calculating a clique cover for Thm 4 and Alg 1:**
> While it is known that obtaining the minimal clique covering is NP-Hard [R1], for many special cases one can find the minimal clique covering in polynomial time, for triangle-free graphs [R2] and perfect graphs [R3]. For general graphs, we can use the algorithm provided in [R4] to obtain an approximate minimal clique covering of size at most 1.25 times the true clique covering, leading to only a constant factor increase in the regret.
>
> [R1] R. M. Karp. Reducibility among combinatorial problems. In Complexity of computer computations, pages 85–103. Springer, 1972.\
> [R2] M. Molloy. The list chromatic number of graphs with small clique number. Journal of Combinatorial Theory, Series B, 134:264–284, 2019.\
> [R3] M. Grötschel, L. Lovász, and A. Schrijver. Stable sets in graphs. In Geometric Algorithms and Combinatorial Optimization, pages 272–303. Springer, 1988.\
> [R4] M. R. Cerioli, L. Faria, T. O. Ferreira, C. A. Martinhon, F. Protti, and B. Reed. Partition into cliques for cubic graphs: Planar case, complexity and approximation. Discrete Applied Mathematics, 156 (12):2270–2278, 2008.\
>
> ---**Improving the presentation of lemmas:** Apart from 2 lemmas that closely follow other lemmas, all the lemmas provided in the paper are substantially different. (Lemma 5 closely follows Lemma3, Lemma 6 closely follows Lemma 2). We will restructure these lemmas to avoid restatements and improve readability.
>
> ---**Result for UCB with delays is known:** Jouliani et al provided results for single-agent UCB with stochastic delays. We provided novel analysis and results for the case where communicated messages are delayed stochastically. This requires bounding the total number of outstanding messages in the network. We are the first ones to provide this analysis. (Analysis is given in Lemma 7.)
>
> ---**References and citations:** We will include missing citations and references in the updated version of the paper.
>
> ---**Algorithm 1 notation $m$ sometimes has i subscript and sometimes doesn’t:** Apologies for the typo! The correct notation is indeed $m_i$, we will fix this in the updated version.
>
> ---**Algorithm 1  definition of $T_i(m)$ and $\Delta_{k}^i(m)$ when $m > 0$:** The update rule for $T_i(m_i)$  is provided in line 9 (at the start of the epoch). Thank you for catching the $\Delta^i_k$ update error. It is as follows. Each agent $i \in \mathcal I$ first computes the mean $\mu^i_k(m)$ of all the cumulative rewards obtained in epoch $m$. Then it computes the value $\mu^i_\star(m) = \max_{k \in [K]}\{\mu^i_k(m) - \frac{1}{16}\Delta^i_k(m-1)\}$. Then it sets the value $\Delta^i_k(m) = \max\{2^{-m}, \mu^i_\star(m) - \mu^i_k(m)\}$.
>
> ---**Equation (59):** We will mention that we first bound the first 3 summands and then the rest is bounded by the integral.
>
> ---**Missing a factor of 3 inside log term in (62):** Thank you for pointing this out. Yes. We are missing a factor of 3. We will include this factor and revise the statements accordingly.
>
> ---**Typos and improving writing:** We will fix typos and revise the paper to improve clarity, readability, and presentation

---

> > ### Comment · Reviewer_9KQ6 · 2021-09-10
> > **response**
> >
> > About the evidence supporting dropping information, thank you for providing the experiment results. However if you can send your local reward you can send your current estimate of the mean, in which case the group regret would become better than your approach and certainly better than dropping useful information.
> >
> > This leads to the other concern. You claim "Analysing the regret of the whole network without considering a clique partitioning leads to poor regret bounds. (large constants multiplying \log T term.) This is due to the fact that messages created by an agent do not propagate to all other agents." I do not believe this to be so simple. Messages created by an agent do not propagate **instantaneously** to all other agents, but if you forward the messages they can be propagated after some time, which would just increase the delay (so your delayed framework one would be able to analyze it) and one would not need several partitions of the graph. Hence, the constant multiplying the $\log T$ would not have the clique number. The price to pay is a greater delay which accounts for a time-independent regret. One could argue that the delay could be too high, but it is not going to be greater than O(clique covering number) (and this just goes to the time independent part of the regret). One could also argue that sending all the information could lead to too many messages, but there are works that just send the current best estimate of the means instead of all the messages. There could be some trade-off here, I'm not saying this is the best thing to do always, but you do not discuss about these other options that do not have constants multiplying log(T) and which are supported by several works already and when each approach would be convenient.
> >
> > For these reasons I am keeping my score

---

### Official Review · Reviewer_huWS · 2021-07-16

**Rating:** 5
**Confidence:** 3

**Summary:**

Final comment: in its current form, I believe this paper is borderline. I went back and forth with it, but having to commit to either a weak acceptance or a weak rejection, considering all comments and the discussion with the other reviewers, I am, unfortunately, picking the latter.
----------

TLDR: The authors study three variants of a cooperative bandit setting with imperfect communication.

Base setting in short: at each round, each agent in a network pulls an arm, sees and gains its corresponding reward, then shares some messages with its neighbors with the goal of minimizing the collective regret.

Variant 1: each message can fail to reach its recipient with probability $1-p$.

Variant 2: messages arrive with a stochastic (but deterministically bounded) delay

Variant 3: the real numbers shared in the messages are corrupted with a small perturbation

The authors introduce and analyze simple algorithms for these three variants.
They also consider the problem with perfect communication, for which they prove an upper and lower bound. The lower bound is improved in a special case but none of them scale with the same problem-dependent constants of the upper bound.

The theoretical results are complemented by some experiments.

**Limitations And Societal Impact:**

The settings considered are not fully fleshed out. I would at least add an Open problem section in which all the open questions are candidly laid out.

**Main Review:**

TLDR: the variants introduced are of some interest but I am not fully convinced that the current state of the paper meets the NeurIPS standards in terms of novelty, techniques, and completeness.

Discussion

The idea of modeling some sort of communication impediment within a cooperative setting is worth exploring. To this end, I suggest adding the reference [1], where the impediment is a constraint in the number of bits of the messages, and look at the references therein.
In particular, I find all three variants studied in this paper to be worthy research directions and of interest to the NeurIPS community.

My main issue is the novelty.
The algorithms are all pretty natural. This would not be a flaw per se but their analysis and the corresponding results are not too different from the most standard techniques and the most expected results that one could think of. This leads me to question the value that this work in (its current state) would add to the community.
On a similar note, this paper seems incomplete. Of the five upper bounds that are presented, only one of them has a corresponding lower bound (that on top of it, is mismatching). Moreover, the paper introduces two protocols for their setting (message-passing and instantaneous reward sharing, 117-118) and it analyzes both of them at the beginning, but then it abruptly drops one of them without giving a real reason why (220-221).
Overall, this feels more like a collection of several loosely related minor results than a coherent paper with a clear message. I would have preferred to see fewer variants (even just one of them) fully fleshed out instead.

Questions/issues on specific sections.

- Section 3.
	- The bound in Thm 1 depends on the clique-covering number. However, [2] suggests that for these types of distributed problems the right scaling is usually with the independence number (which is always smaller). Do you believe your analysis could be refined or do you have an argument for why the correct constant is the clique-covering number?
	- Remark 1 is fine but it would be preferable to have a proper lower bound, that matches the dependence on p and $\bar \chi$ of Thm 1
	- In Remark 2, the tuning of $p_i$ depends on $d_{\min}(G)$, meaning that the nodes need some knowledge of the global structure of the network. This is an undesirable requirement in cooperative learning. Is there a way to keep the same bound but only require the agents to have local knowledge (i.e., only about their neighborhoods)?
	- Thm 2. Same question of Thm 1 on $\bar \chi$ vs $\alpha$.
	- Remark 3. Similar remark and question as for Remarks 1 and 2.
- Section 4
	- Why dropping message-passing? Inconsistent approach.
	- Applications of delayed analysis in cooperation already appeared in recent work. I suggest adding a reference to [3] and look at the references therein.
	- Remark 4. See Remark 1.
- Section 5
	- Remark 5. See Remark 1.
- Section 6
	- Thm 5. Same remark as per Thm 1. Should the bound depend on $\bar \chi$ or $\alpha$? Also, $h$ depends on $d_i$. Who is $i$? Does it hold for any agent $i$? If so, there could be a minimum over agents.
	- Thm 6. The first bound depends on a constant $\widetilde d$ that is a function of terms $\bar d_{=i}$ that were never introduced, so it is not clear what that means. Still, it does not depend on $\bar \chi$, showing that at least one between the upper and this lower bound is loose. In the second bound, I finally see the form I was expecting. If I had to make an educated guess, I would say that this is the correct rate and I would recommend trying to extend this result to the general case and finding an algorithm that matches it (or tightening the analysis of Delayed MP-UCB, if possible).


[1] Della Vecchia et al. An Efficient Algorithm for Cooperative Semi-Bandits, 2021.

[2] Cesa-Bianchi et al. Delay and cooperation in nonstochastic bandits, 2019.

[3] Hsieh et al. Multi-agent online optimization with delays: Asynchronicity, adaptivity, and optimism, 2020.


Minor comments

The writing is not always great. In particular:

- 109. As presented, $\Delta_2$ could be equal to $\Delta_1$. If this is the case, there are a bunch of divisions by $0$ scattered throughout the paper. Unless you assume that there is a unique maximizer $\mu_1$, you should clean up all the expressions in which the sum over $\Delta_i$ appears at the denominator (it is probably just a matter of summing only on suboptimal arms).
- 111. I strongly suggest avoiding the use of f and g (this choice makes things gratuitously hard to read), replacing them with reasonably looking big-O's in the theorems where they appear.
- 111. (not-so-minor comment) In many places in the paper, $o(\cdot)$ is used where it should really be $O(\cdot)$.
- 111. (not-so-minor comment) What is the role of $\xi$? Based on your upper bounds and the definition of $Q_k^i$ in Fig. 2, it seems that you would be better off by always picking $\xi$ as small as possible. Is that so? This way, you could get rid of a parameter.
- 131. size of the smallest possible covering -> size of the smallest possible clique cover
- 165. (this relates to all other algorithms as well) The clarity would greatly improve if each algorithm had its float rather than having a generic one plus added text mixed with the rest of the narrative.
- 221. future work. -> future work).
- 223. Vernade et al. (2017). -> Vernade et al. (2017)).
- 249. You call your algorithm RCL-AC but point to a float where it is called CHARM. Pick one.
- Algorithm 1. Why do you need the huge constant 1024?
- 290. I would consider stating Thm 4 like this since it cannot really be used with the smallest dominating set.
- 295. delayed(MP) -> Delayed(MP). Also, the font for "Delayed" is wrong.
- 296. I highly recommend restating this theorem in a cleaner, more explicit manner.

**Time Spent Reviewing:**

10

---

> ### Author Response · Authors · 2021-08-11
> **Author Response**
>
> Thank you for the really constructive feedback! Your feedback has definitely improved our draft, and we are answering the questions below.
>
> ---**Missing references:** Thank you for pointing out missing references. We will include these references in the revised version of the paper.
>
> ---**Clique covering number vs independence number:** We would like to clarify that all our algorithms do not scale with the clique covering number. RCL-AC (Section 5) in fact has a regret that scales with the domination number, which is always smaller than the independence number (and consequently the clique-covering number).  While [R2] does demonstrate a scaling with the independence number, that scaling has not been demonstrated to be optimal (See Section 7, point 2 of [R2]), and moreover, can be improved to the dominating number in certain (Section 5).
>
> We can provide more insight into this dependence. Observe that in the first two settings, the algorithms proposed are decentralized and individually consistent (i.e, agents do not mimic other agents), and the messages themselves are of $\mathcal O(\log(NK)$ bits, which leads to the weaker clique-covering bound. Alternatively, Section 5 introduces an algorithm that is not completely decentralized (as some agents mimic other agents), lowering the bound. This technique can also be applied to the first two settings to reduce the regret to the domination number, but the algorithms are not decentralized in that case. Alternatively, one can obtain an independence number bound by using messages of length $\mathcal O(K\log(NK))$ bits by replicating this centralized behavior in a decentralized manner. This approach has been presented in [R1] (see Section 6.3) and can potentially be extended to the corrupted case as well. Note that the approach presented by [R2] also utilizes $\mathcal O(K \log(NK))$ bits per message, in contrast to the approach in Sections 3 and 4. We present only the first approach for simplicity but we will include this remark in the final version.
>
> In the absence of any constraint on the policy, the lower bounds are difficult to quantify and generally cannot be improved beyond $\Omega(K\log T)$ due to arbitrary communication protocols. For example, consider a policy where agents send messages of $\mathcal O(NK)$ size, where the message contains the latest reward and arm pull for all $N$ agents that the sending agent has seen. This can be demonstrated to diffuse information entirely across the network (albeit with a maximum delay of $d\_\star(G)$), and can hence be considered as one decision-making unit to obtain an $\mathcal O(K\log T + KNd\_\star(G))$ regret regardless of $G$. However, $\mathcal O(NK)$-sized messages are infeasible in practice, and hence a latent assumption that is employed in this setting is that messages can be of $\mathcal O(\log(NK))$ bits.
>
> Under this assumption, a near-optimal _problem dependent_ asymptotic lower bound has been established for reward-sharing in the work of [R3], which scales as $\Omega(K\alpha(G_2)\log T)$ for a class of non-altruistic and individually consistent policies (NAIC), i.e, agent policies that do not mimic other agents or act to minimize global regret at the expense of individual regret (the technical version of this statement can be found in [R3], Section 4). This result can directly be extended to message-passing with delays to give a corresponding lower bound of $\Omega(K\alpha(G_{\gamma+1})\log T)$ for any $\gamma < d_\star(G)$ and $\Omega(K\log T)$ for $\gamma = d_\star(G)$. While we mention this briefly in Section 6 (after Remark 7), we omit a full proof for brevity, which we can update in the final version. Observe that even in the sparsest connected graph, i.e., a line graph, we have that $\alpha(G_{\gamma+1}) = \frac{N}{\gamma+1}$ whereas the clique covering number in the upper bound $\bar\chi(G_\gamma) = \frac{N}{\gamma}$ (ignore rounding issues for now). This shows that the upper bound is within a constant $1+\frac{1}{\gamma} \leq 2$ of the lower bound, independent of $N$, even in the worst case. Now, this bound holds only for policies that are NAIC, which the algorithms presented in Sections 3 and 4 satisfy. Note that the algorithm in Section 5 is not NAIC and hence does not satisfy the bound.
>
>
> [R1] Dubey, A. and Pentland, A. Cooperative multi-agent bandits with heavy tails. International Conference on Machine Learning. PMLR, 2020.\
> [R2] Cesa-Bianchi et al. Delay and cooperation in nonstochastic bandits, 2019.\
> [R3] Kolla, R. K., Jagannathan, K., & Gopalan, A. (2018). Collaborative learning of stochastic bandits over a social network. IEEE/ACM Transactions on Networking, 26(4), 1782-1795.
>
> ---**Obtaining a lower bound for Theorems 1, 2, 3:** As mentioned earlier it is difficult to provide a tight lower bound for this case due to the complexity of the joint probability distribution of the sampling process of agents. Deriving tight lower bounds for multi-agent cooperative algorithms is an active area of research.
>
> ---**Knowing $d_{\min}(G)$ in Remark 2:** We agree that knowing the minimum degree of the network is knowing a global property of the network which is not locally available. We can propose two methods for relaxing this assumption. 1.) Introducing a communication phase that allows agents to figure out the minimum degree by passing messages about the degree of agents. This requires at most the number of communication rounds equal to the diameter of the graph. During this period agents can use $p_i=\frac{1}{d_i},$ which also effectively reduces the information bias across the network. 2.) Instead of $d_{\min}(G)$, agents can use the minimum degree of their neighbors. We believe it is reasonable to assume that agents know their own degree and the degree of their neighbors, reducing the information bias in the network locally.
>
> ---**Knowing $d_{\min}(G_{\gamma})$ in Remark 3:** It is clear that the first method mentioned for relaxing this assumption in the previous section is applicable in this case also. However, the second method is less applicable since it is unreasonable to assume that agents can know the degree of their $\gamma$ hop neighbor. However, this method can be applied if agents use a communication method to pass messages about their id and degree (which can be done in $\gamma$ rounds).
>
> ---**Why dropping message-passing?:** Note that in a general communication network $G$ there can be more than one path between any given pair of agents. Thus agents can receive messages from different paths. When there is a stochastic delay associated with passing messages it is difficult to analyze the expected delay of a message. When this is combined with the stochasticity associated with the joint sampling process of agents, regret analysis becomes significantly difficult. One way to simplify the analysis is considering that there is always a fixed path to send messages between a given pair of agents. Although this allows us to provide meaningful regret bounds this results in increasing group regret compared to the case where agents can use message passing without committing to a specific path. Thus we left proposing a suitable algorithm that uses message passing under stochastic communication delays as future work.
>
> ---**Reference [3]:** Thank you for pointing out this reference. We will add this reference in the revised version of the paper. They indeed consider the delay analysis in cooperation. However, they do not consider the effect of delays in cooperation under exploration and exploitation, which is the focus of our work.
>
> ---**Theorem 6:**
> We apologize for the missing notation. The first part of Theorem 6 refers to $\bar d_{=i}$ which equals the average number of neighbors at distance $i$ in the graph. In Remark 8, we present some comparisons on the lower bound for agnostic policies and observe that the lower bound is within constant factors for several graph families. For example, for line graphs, our lower bound $\alpha_\star(G_\gamma) = \frac{N}{1+\gamma}$ whereas the obtained bounds are $\mathcal O(\frac{N}{\gamma})$. We believe that the lower bound is tight for these classes of policies and is not an artifact of our analysis. Regarding the upper bound, however, we believe that
>
> ---**Avoiding divisions by zero:** Thank you for this remark, indeed our analysis (and bounds) should be rewritten as $\sum_{k: \Delta_k > 0}$ to avoid these errors. We will update the final version of the paper correcting these issues.
>
> ---**notations o(.) and O(.):** Thanks for pointing this out. Line 111 is $o(1).$ Line 112 should be $O(.).$ We will update the  $o(.)$ and $O(.)$ notations in the revised version of the paper.
>
> ---**The role of ξ:** The role of $\xi$ is regulating the trade-off between exploring and exploiting by regulating the weighted average between estimated mean and the confidence bound term. We used the constant in front of $\log t$  in the confidence bound as $2(\xi+1)$ for algebraic convenience.
>
> ---**The reason for the large constant 1024 in Algorithm 1:** Algorithm 1 involves an epoch-based approach to eliminate arms, where the epoch size increases exponentially. We formulate the epoch size as an appropriate power of $2$, where our analysis recovers a constant $2^{10}$ by balancing the probability of eliminating a wrong arm in the first (and successive) epochs vs. epoch length. We can reduce the length of the epoch and use a smaller constant at an increase in the (expected) regret by picking the wrong arm. We will include a discussion in the updated paper.
>
> ---**Adding a section on open problems:** One open problem is providing matching lower bounds. Another open problem is providing an algorithm that uses message passing under stochastic communication delays. We will include a section on open problems in the revised version of the paper.
>
> ---**Typos and improving writing:** We will fix typos and revise the paper to improve clarity, readability and presentation

---

### Official Review · Reviewer_CiVs · 2021-07-21

**Rating:** 6
**Confidence:** 3

**Summary:**

The paper studies the cooperative bandit problem. Players can communicate with each other, but the communication can be imperfect.
The authors consider three possible imperfect communication--link failures, stochastic delay, and adversarial corruption. For each case, they give algorithms and provide performance guarantees. For the case of perfect communication, they develop an improved algorithm for the case of stochastic delays and give a minimax lower bound.

**Limitations And Societal Impact:**

Yes

**Main Review:**

Communication constraints arise naturally in many real-life applications of cooperative bandits. The study of communication constraints in cooperative bandits is timely and relevant. However, the aspects like delayed and adversarial corruption are looking into separately in bandit literature. Based on this the idea and techniques proposed in this paper look incremental. I have the following questions.

1) Is it possible to show that the regret bound in Thm. 1 is always better than the regret incurred in the case on no-communication setup for all possible $p$ and $p_i$s
2)  Line 192: Probilites  $(\marx_{i\leq N} p_i)p$ and $\sum_{i\leq N} (1-p_i*p)$ do not add to one. Why they are treated as probabilities for perfect communication and no-communication regret. If their sum exceeds one, what is common between perfect communication and no-communication regret, and if their sum does not exceed one, which event is missing?


------
Post rebuttal:

I looked into their rebuttals. The authors partly answered my 2nd question, but the answer to the first question is unclear. From their arguments, I do not see how the group performance for any communication probability p>0 is better than the no-communication case.
Post rebuttal I still feel that this paper is incremental and hence retain my score.








**Time Spent Reviewing:**

7

---

> ### Author Response · Authors · 2021-08-11
> **Author Response**
>
> Thank you for the constructive feedback! In the following, we address your concerns individually.
>
> ---**For all possible $p$ regret bound in Thm 1 is better than no-communication regret:** The result given in Theorem 1 can be slightly improved by replacing $\bar{\chi}(G)(\max_{i\leq N}p_i)$ with $\sum_{\mathcal{C}\in \mathfrak{C}}(\max_{i\leq \mathcal{C}}p_i),$ where $\mathfrak{C}$ is a non overlapping clique covering. (This result follows from equation 17.) Then we obtain $\sum_{i=1}^N(1-p_ip)+\sum_{\mathcal{C}\in \mathfrak{C}}(\max_{i\leq \mathcal{C}}p_i)p=N-p(\sum_{i=1}^Np_i-\sum_{\mathcal{C}\in \mathfrak{C}}(\max_{i\leq \mathcal{C}}p_i)).$ Since the clique covering is non overlapping  the results show that agents obtain improved group performance for any communication probability $p>0$ for any non trivial graph compared to no communication case where each agent learns by itself.
>
> ---**Elaborating Line 192:** Line 193 has a typo. It should be $\frac{1}{N}\sum_{i=1}^N(1-p_ip)$ instead of $\sum_{i=1}^N(1-p_ip).$ Note that when we substitute $p_i=\widehat{p},\forall i$ we obtain $sum_{i=1}^N(1-p_ip) + \bar{\chi}(G)(\max_{i\leq N}p_ip) =N(1-p\widehat{p})+\bar{\chi}(G)p\widehat{p}.$ This implies that with probability $1-p\widehat{p}$ regret scales as $N$ agents in isolation and with probability $p\widehat{p}$ regret scales as a perfect communication network $G.$ When agents have different message discarding probabilities $p_i$ we that $\frac{1}{N}\sum_{i=1}^N(1-p_ip)+\max_{i\leq N}p_ip$ is greater than 1. This does not arise due to a common event between perfect communication and no communication. This simply arise because we are using an upper bound for $\sum_{i \in \mathcal{C}} p_i p \mathbb{E}(n^{i}\_k(\tau\_{k, \mathcal{C}}))$ (last term in equation 17). We will clarify this remark in the updated version.

---

### Author Response · Authors · 2021-08-11
**Common Responses**

We would like to thank all reviewers for their valuable feedback. Here we address the common issue of novelty about our approaches in more detail.

In this paper, we consider three types of communication imperfections and we discuss the novelty and non-incremental nature of results in each imperfection as follows. Despite the simplicity of our ideas, theoretical analysis poses significant challenges. In addition, we discuss the novel and non-incremental nature of the theoretical analysis and results presented for perfect communication.

Link failures: Our work is the first work to provide general theoretical results for the case with probabilistic communication over general communication networks. Providing theoretical results for general network graphs with probabilistic communication poses two significant theoretical challenges. 1.) Providing a tail probability bound for the case where agents communicate probabilistically (see Lemma 3 and Lemma 5) and 2.) decomposing the regret into high probability events that can be bounded depending on the network structure and low probability event that can be upper bounded using tail probabilities (see Lemma 2 and Lemma 6).

Random delays: To the best of our knowledge our work is the first work to study random communication delays in cooperative stochastic bandits. We provide novel theoretical analysis for this setting (see Lemma 7) by proving that the group regret can be decomposed into the summation of regret under perfect communication and a term that is proportional to the expected number of outstanding messages in the network.

Corruptions: We present a novel combination of arm elimination and UCB exploration for the adversarially corrupted communication setting. Our analysis involves 2 novel components: first, we provide a new analysis of epoch-based arm elimination under communication. This requires a new argument to bound the probability of the bad arm being picked (Lemma 10). Next, we provide a new algorithm to handle delays in the communication of arms: to our knowledge, arm elimination in a delayed feedback setting has not been explored before. We demonstrate that by running local estimation algorithms in between arm elimination epochs, one can obtain only a $\mathcal O(N\log\log T)$ additional increase in regret

Lower bound: Our lower bounds for minimax rates are the first of their kind in this setting and for many graph families (linear, circular, regular, etc.) are within a constant factor of the upper bound presented in our algorithms (see Remark 8).

---

> ### Comment · Area_Chair_UFXc · 2021-08-24
> **Reference**
>
> > to our knowledge, arm elimination in a delayed feedback setting has not been explored before.
>
> Take a look at Tal Lancewicki, Shahar Segal, Tomer Koren, and Yishay Mansour, “Stochastic Multi-Armed Bandits with Unrestricted Delay Distributions”, ICML-2021. Although, personally, I am not a big fan of arm elimination. Algorithms that do not resort to arm elimination are much more elegant and versatile.

---

> > ### Author Response · Authors · 2021-08-30
> > **Thank you for the reference!**
> >
> > Thank you for the reference! It indeed is very relevant, however since it is so recent - it appeared on Arxiv on June 4 (i.e., six days after the submission deadline), our remark was valid at the time of submission. We will update the paper to highlight differences with this paper, however a quick glance suggests that our problem setting in itself is markedly different, where we have (a) multiple agents, and (b) corrupted rewards, in contrast to the setting discussed in the paper. We would also like to remark that only one algorithm we present is based on arm elimination (i.e., in the presence of adversarial corruptions).

---

### Decision · Program_Chairs · 2021-09-28

**Decision:**

Accept (Poster)

**Comment:**

The paper failed to generate sufficient support among the reviewers, who consider the results to be incremental.

**Consistency Experiment:**

NeurIPS has a long history of experimentation. In 2014, NeurIPS ran an experiment in which 10% of submissions were reviewed by two independent committees to quantify the randomness in the review process. This year, we repeated a variant of this experiment to see how the quality of the review process has changed over time.  This paper was part of the experiment and was therefore assigned to two committees (consisting of reviewers, an Area Chair, and a Senior Area Chair) that reached independent decisions.  If both committees made the same recommendation, this recommendation was followed. If a single committee recommended acceptance, the paper was accepted (with the exception of a few cases in which the other committee identified what we considered a fatal flaw, e.g., an error in a key result).

This copy’s committee reached the following decision: **Reject**

The other committee assigned to the paper recommended **Accept (Poster)**.  You can find the other set of reviews, along with any follow up discussion with the authors here:
https://openreview.net/forum?id=PmJVah9D8B